# LOOKAHEAD TREE-BASED ROLLOUTS FOR ENHANCED TRAJECTORY-LEVEL EXPLORATION IN REINFORCEMENT LEARNING WITH VERIFIABLE REWARDS

**Shangyu Xing**[1]    **Siyuan Wang**[2*]    **Chenyuan Yang**[3]    **Xinyu Dai**[1]    **Xiang Ren**[2*]
[1] Nanjing University    [2] University of Southern California    [3] Fudan University

## ABSTRACT

Reinforcement Learning with Verifiable Rewards (RLVR), particularly with algorithms like Group Relative Policy Optimization (GRPO), has proven highly effective in enhancing the reasoning capabilities of large language models. However, a critical bottleneck in current pipelines lies in the limited diversity of sampled trajectories during group rollouts. Homogeneous trajectories and their associated rewards would diminish the return signals for policy updates, thereby hindering effective policy learning. This lack of diversity stems primarily from token-level stochastic sampling, where local variations are likely to collapse into near-identical reasoning paths. To address this limitation, we propose Lookahead Tree-Based Rollouts (LATR), a novel rollout strategy designed to explicitly promotes trajectory-level diversity by enforcing branching into different candidate tokens likely to yield distinct continuations. Specifically, LATR iteratively operates in three stages: (1) branching at high-uncertainty generation steps, (2) performing lookahead simulation for each new branch, and (3) pruning branches that exhibits prolonged similarity during simulation. Compared with Stochastic Sampling, LATR accelerates policy learning by 131% and improves final pass@1 performance by 4.2% on both GRPO and Dynamic sAmpling Policy Optimization (DAPO) algorithms across different reasoning tasks. Our code and data are available at `https://github.com/starreeze/latr`.

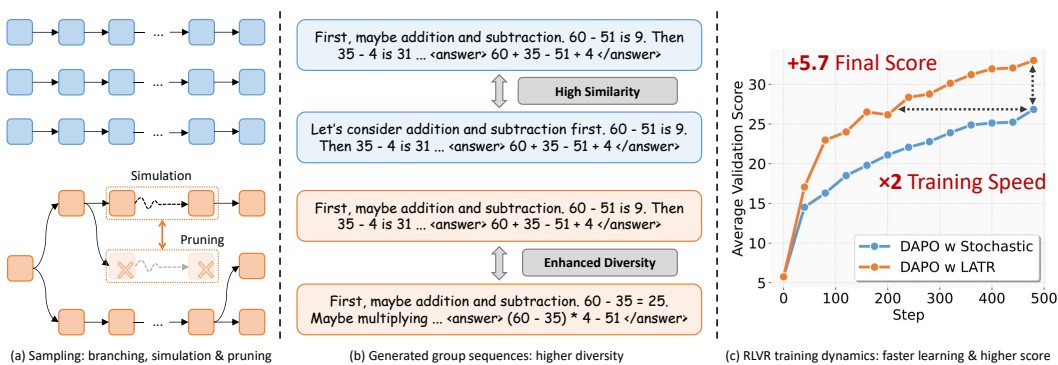

Figure 1: Comparison of conventional token-level **stochastic sampling** and our proposed method **LATR** on sampling process, rollout sequence diversity, and performance on DAPO Math dataset.

# 1 INTRODUCTION

Reinforcement learning with verifiable rewards (RLVR) has emerged as a powerful paradigm for enhancing the reasoning capabilities of large language models recently (DeepSeek-AI et al., 2025; Yang et al., 2025; OpenAI, 2025). By leveraging sequence rollouts and updating policies according

---

*Correspondence to `sw_641@usc.edu` and `xiangren@usc.edu`.

to appropriate rewards, RLVR can significantly improve performance across diverse reasoning tasks, including mathematical problem solving, code generation, and multi-step logical deduction (Pan et al., 2025). Algorithms such as Group Relative Policy Optimization (GRPO) (Shao et al., 2024) have become central to this approach, enabling stable model training through in-group trajectory comparisons to learn from high-quality responses while penalizing low-rewarded ones.

A key challenge in these methods lies in the limited diversity of trajectories sampled during the rollout phase (Wang et al., 2025; Zhu et al., 2025). When trajectories within a group exhibit high similarity, the estimated relative advantage and learning signals tend to diminish. As a result, the policy updates become less informative, ultimately hindering the effective scaling. Recent efforts have sought to mitigate this issue through various approaches, including increasing sampling temperature (Liu et al., 2025) and dynamically filtering out groups with highly similar samples (Yu et al., 2025). However, the former focuses on token-level variation without ensuring trajectory-level divergence, while the latter relies on post hoc filtering that provides only limited within-group diversity at the cost of excessive over-generation. Both methods therefore yield only modest improvements in rollout diversity under a constrained generation budget.

We argue that such diversity limitation stems from the predominate reliance on token-level stochastic sampling strategies, where each sequence in a group is generated independently by sampling tokens from the model's output distribution at each decoding step. While simple and widely adopted, this approach ignores the contrast among sequences within the group and fails to enforce distinction or complementarity at the trajectory level, thus exhibiting an inherently myopic limitation. Specifically, token-level variations typically occur without lookahead ability, making local deviations (e.g., substituting "compute" with "calculate") easily collapse back into nearly identical reasoning paths, leading to redundant exploration and diminishing returns.

To address these limitations, we propose **L**ook**a**head **T**ree-Based **R**ollout (LATR), a strategy designed to explicitly promote trajectory-level diversity within a group by maintaining rollouts in a tree structure. At token positions with high model uncertainty, LATR enforces branching into different candidate tokens that are highly likely to yield distinct continuations. To guarantee that each selected candidate token can lead to a different reasoning path, LATR performs lookahead simulation by continuing generation for a fixed length, and removes those candidates failing to diverge from others. This branching, simulation and pruning procedure is iteratively repeated until the target number of rollouts is reached, after which all surviving partial sequences continue to be extended in parallel under standard stochastic sampling. This ensures that the generated trajectories are reasonably distinct from each other, thereby enriching the in-group rollout diversity.

We apply LATR strategy to both GRPO and DAPO algorithms and evaluate across 5 datasets involving mathematical and logical reasoning. Our experiments demonstrate that LATR consistently accelerates policy learning by an average of 131%, while simultaneously improving final task performance of pass@1 by averagely 4.2% across different tasks. Our contributions are summarized as follows:

1) We introduce a novel tree-based rollout algorithm LATR that explicitly optimizes for trajectory-level diversity, which can be integrated seamlessly into any policy update algorithms.
2) We provide extensive empirical validation across tasks and training configurations, demonstrating consistent improvements over existing sampling strategies in RLVR pipelines.

## 2 PRELIMINARY

We adopt Group Relative Policy Optimization (GRPO) (Shao et al., 2024) as the foundational RL algorithm for policy refinement. Unlike Proximal Policy Optimization (PPO) (Schulman et al., 2017), which relies on a learned value function to estimate advantages, GRPO eschews explicit value modeling and instead computes advantages directly from group-relative rewards. This design simplifies training dynamics and enhances stability in reward-sparse or high-variance environments.

Each training iteration in GRPO consists of two phases: (1) *rollout*, where multiple candidate responses are sampled per prompt, and (2) *policy update*, where the policy is optimized using group-normalized advantages and a clipped surrogate objective.

**Rollout.** Given a prompt $p$ drawn from the dataset $\mathcal{D}$, a group of $k$ candidate sequences $\{s_i\}_{i=1}^k$ are generated via autoregressive sampling from policy $\pi_\theta$. Each sequence is constructed token-by-token

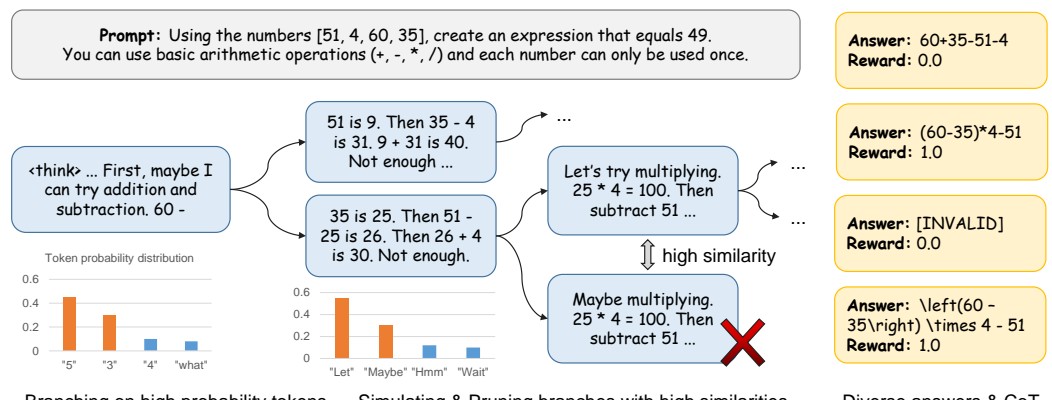

Figure 2: An overview of LATR. A dynamic search tree is built by branching on model uncertainty, simulating and pruning similar branches, resulting in diverse answers and reasoning paths.

through stochastic sampling from the model's predicted next-token distribution. This process is identical to inference-time generation.

Formally, let $S_l$ denote the multiset of partial sequences of length $l$ generated for prompt $p$. The rollout process is recursively defined as:

$$S_0 = \{\underbrace{\epsilon, \epsilon, \ldots, \epsilon}_{k}\}, \quad S_{l+1} = \bigcup_{s \in S_l} \{s \oplus t \mid t \sim \pi_\theta(\cdot \mid p \oplus s)\}, \tag{1}$$

where $\epsilon$ is the empty sequence, $\oplus$ denotes token concatenation, and sampling terminates when all sequences reach an end-of-sequence token or a maximum length $n$. The final output is the group $S = \{s_1, \ldots, s_k\}$. This group-based sampling enables direct comparison of responses under the same prompt, forming the basis for relative advantage estimation.

**Policy Update.** Policy update aims to refine policy $\pi$ by maximizing expected cumulative rewards. Similar to PPO, GRPO adopts a clipped objective, together with a directly imposed KL penalty term:

$$\mathcal{J}_{\text{GRPO}}(\theta) = \mathbb{E}_{p \sim \mathcal{D}, \{s_i\}_{i=1}^k \sim \pi_{\theta_{\text{old}}}(\cdot|p)}$$

$$\left[ \frac{1}{k} \sum_{i=1}^k \frac{1}{|s_i|} \sum_{l=1}^{|s_i|} \left( \min \left( r_{i,l}(\theta)\hat{A}_{i,l}, \ \text{clip}\left(r_{i,l}(\theta), 1-\varepsilon, 1+\varepsilon\right)\hat{A}_{i,l} \right) - \beta D_{\text{KL}}(\pi_\theta||\pi_{\text{ref}}) \right) \right], \tag{2}$$

where the advantage $\hat{A}$ is calculated by normalizing the group-level rewards $\{R_i\}_{i=1}^k$, and the ratio $r$ compares the likelihood of token $s_{i,l}$ under the current and old policies:

$$\hat{A}_{i,l} = \frac{R_i - \text{mean}(\{R_i\}_{i=1}^k)}{\text{std}(\{R_i\}_{i=1}^k)}, \quad r_{i,l}(\theta) = \frac{\pi_\theta(s_{i,l} \mid p, s_{i,<l})}{\pi_{\theta_{\text{old}}}(s_{i,l} \mid p, s_{i,<l})}. \tag{3}$$

**Variations of GRPO.** Building upon this, DAPO (Yu et al., 2025) improves GRPO in several aspects. In rollout stage, DAPO oversamples data batches and filters out groups with identical rewards. If the retained groups are insufficient to fill a batch, additional rollouts are iteratively sampled. This mechanism trades computational efficiency for higher response diversity and more informative gradients. In policy update stage, DAPO addresses GRPO's limitations in long-form generation tasks by implementing token-level loss calculation to mitigate length bias, and employs decoupled clipping without RL penalty to encourage exploration. Formally, the objective is

$$\mathcal{J}_{\text{DAPO}}(\theta) = \mathbb{E}_{p \sim \mathcal{D}, \{s_i\}_{i=1}^k \sim \pi_{\theta_{\text{old}}}(\cdot|p)}$$

$$\left[ \frac{1}{\sum_{i=1}^k |s_i|} \sum_{i=1}^k \sum_{l=1}^{|s_i|} \min \left( r_{i,l}(\theta)\hat{A}_{i,l}, \ \text{clip}\left(r_{i,l}(\theta), 1-\varepsilon_{\text{low}}, 1+\varepsilon_{\text{high}}\right)\hat{A}_{i,l} \right) \right]. \tag{4}$$

These enhancements make DAPO a more robust and effective algorithm for complex reasoning tasks.

# 3   LOOKAHEAD TREE-BASED ROLLOUT

To address the limited diversity of conventional token-level sampling during the rollout phase, we introduce **L**ook**a**head **T**ree-Based **R**ollout (LATR), a structured exploration strategy inspired by Monte Carlo Tree Search (Silver et al., 2016). LATR achieves diverse trajectory generation by enforcing branching at candidate tokens that are highly likely to yield distinct continuations.

Specifically, LATR operates through three iterative stages: (1) *Branching*, which creates new trajectories at token positions with high model uncertainty; (2) *Lookahead Simulation*, where the new branch is extended for a fixed lookahead window of $r$ tokens; and (3) *Pruning*, where simulated sequences that fail to diverge from others are removed. This process repeats until the target number of rollouts is reached, ensuring their diversity. We provide the complete algorithm in Algorithm 1 and an illustration in Figure 2.

## 3.1   BRANCHING

LATR begins with a root node corresponding to the input prompt. At each generation step $l$, every active branch is extended by its highest-probability token to ensure progress along the most likely trajectory. These branches are regraded as parent branches. Simultaneously, if other candidate tokens satisfy both the absolute probability threshold $\tau_{\text{abs}}$ and the relative probability threshold $\tau_{\text{rel}}$, new child branches are instantiated. This dual-threshold mechanism targets reasoning crossroads where the model is genuinely uncertain between semantically distinct continuations, while preventing the branches diverging too far from the policy distribution. Branching allows LATR to maintain multiple distinct reasoning paths in parallel, significantly increasing the probability of discovering high-quality, diverse solutions.

---

**Algorithm 1** Lookahead Tree-Based Rollouts

**Require:** Policy model $\pi$, rollout number $k$, prompt $p$, absolute branching threshold $\tau_{\text{abs}}$, relative threshold $\tau_{\text{rel}}$, pruning threshold $\tau_{\text{ed}}$, lookahead step $r$, max length $n$.

**Ensure:** Set of rollouts $S = \{s_1, \ldots, s_k\}$.
1:  Initialize $S \leftarrow \{\epsilon\}$         ▷ Single root branch
2:  **for** $l = 1$ to $n$ **do**
3:     $S_{\text{next}} \leftarrow \emptyset$
4:
5:     {- - - - - - Branching logic - - - - - -}
6:     **for** branch $s_i \in S$ **do**
7:        $\mathcal{P}_i \leftarrow \pi(\cdot \mid p \oplus s_i)$   ▷ Prob distribution
8:        $c_i^\star \leftarrow \arg\max_c \mathcal{P}_i[c]$    ▷ Top candidate
9:        $s_i^{\text{extend}} \leftarrow s_i \oplus c_i^\star$         ▷ Extend main
10:       $S_{\text{next}} \leftarrow S_{\text{next}} \cup \{s_i^{\text{extend}}\}$
11:       $\mathcal{C}_i \leftarrow \{c \neq c_i^\star \mid \mathcal{P}_i[c] > \tau_{\text{abs}}$ and $\mathcal{P}_i[c_i^\star] - \mathcal{P}_i[c] < \tau_{\text{rel}}\}$
12:       **for** $c \in \mathcal{C}_i$ **do**
13:          **if** $|S_{\text{next}}| < k$ **then**
14:             $s_{\text{new}} \leftarrow s_i \oplus c$          ▷ New branch
15:             $s_{\text{new}}.\text{parent} \leftarrow s_i$
16:             $s_{\text{new}}.\text{birth} \leftarrow l$
17:             $S_{\text{next}} \leftarrow S_{\text{next}} \cup \{s_{\text{new}}\}$
18:          **end if**
19:       **end for**
20:    **end for**
21:
22:    {- - - - - - Pruning logic - - - - - -}
23:    **for** $s_j \in S_{\text{next}}$ with $s_j.\text{birth} = l - r$ **do**
24:       **if** EditDist$(s_j, s_j.\text{parent}) < \tau_{\text{ed}}$ **then**
25:          Remove $s_j$ with its descendants
26:       **end if**
27:    **end for**
28:
29:    $S \leftarrow S_{\text{next}}$
30: **end for**
31: **return** Pad $S$ to exactly $k$ sequences

---

Formally, let $S_l$ denote the set of active branches at step $l$, and for each branch $s \in S_l$, let $\mathcal{P}_s$ denote its next-token distribution, the most likely token $c_s^\star = \arg\max_c \mathcal{P}_s[c]$, and $\mathcal{C}_s$ is the set of all remaining candidates excluding $c_s^\star$. A new child branch $s \oplus c$ is created if:

$$\mathcal{P}_s[c] > \tau_{\text{abs}} \quad \text{and} \quad \mathcal{P}_s[c_s^\star] - \mathcal{P}_s[c] < \tau_{\text{rel}}. \tag{5}$$

The set of branches after expansion is the union of the parent branches with their new children:

$$S_l' = \bigcup_{s \in S_l} \left(\{s \oplus c_s^\star\} \cup \{s \oplus c \mid c \in \mathcal{C}_s, \text{ conditions of (5) hold}, |S_l'| < k\}\right). \tag{6}$$

If the rollout budget $k$ is reached, candidate branches are prioritized by descending probability $\mathcal{P}_s[c]$. This ensures that more plausible alternatives are more likely to be explored.

## 3.2 SIMULATION & PRUNING

While the above branching strategy effectively enables structured parallel exploration, it faces two challenges: (1) unconstrained branching leads to exponential growth, quickly exhausting the rollout budget and limiting exploration sequentially; (2) branches started from token-level variations easily collapse back into nearly identical reasoning paths, struggling to ensure trajectory-level diversity.

To address these issues, LATR incorporates a lookahead simulation and pruning phase. After branching, each new trajectory continues generation for a fixed lookahead window of $r$ tokens. These continuations are then evaluated for divergence using normalized edit distance, and branches exhibiting insufficient divergence from their parents are pruned.

Specifically, at each step $l$, LATR identifies all branches $s$ created at step $l - r$ and computes the normalized edit distance over their most recent $r$ tokens relative to their parents' corresponding segment. If the distance falls below a threshold $\tau_{\text{ed}}$, the branch and all its descendants are removed:

$$S_l^{\text{prune}} = \{s \mid s \in S_l', \ s.\text{birth} = l - r, \ \text{EditDist}(s[-r:], s.\text{parent}[-r:]) < \tau_{\text{ed}}\}, \quad (7)$$

$$S_{l+1} = \{s \mid s \in S_l', \ s \notin S_l^{\text{prune}}\}, \quad (8)$$

where EditDist() indicates normalized edit distance, i.e., the Levenshtein distance between token ID sequences, divided by sequence length. This ensures that only branches exhibiting meaningful divergence within the lookahead window are preserved. We also explore similarity measures other than edit distance in Appendix C.2, and find that their performance are very close. Notably, LATR is backtracking-free, so the number of forward passes required by a group rollout is bounded by $O(nk)$, where $k$ is the rollout number (tree width) and $n$ is the maximum completion length (tree depth).

Through lookahead simulation and pruning, LATR preserves only diverse branches that are more likely to yield distinct reasoning paths. The final output consists of $k$ surviving branches, padded if necessary to meet the rollout number requirements. The entire procedure is compatible with any autoregressive language model and can be integrated seamlessly into existing policy update algorithms and RLVR frameworks without modifications.

## 3.3 FURTHER OPTIMIZATIONS

**Early Stopping.** When the tree width reaches the rollout number $k$, LATR has already produced $k$ sequences that are likely to lead to diverse reasoning paths. At this point, the generation process is switched to standard stochastic sampling for all remaining steps. This allows surviving branches to continue exploring the solution space stochastically while maintaining the diversity benefits from LATR. Analyses on the stopping length is in Appendix C.5.

**Hybrid Rollout for RL Training.** While LATR excels at promoting diverse exploration during RL training, its explicit divergence objective can create a mismatch with test-time behavior. At real-world inference, models typically generate a single trajectory using greedy or stochastic decoding, prioritizing correctness and coherence over diversity. However, policy updates with LATR tries to maximize the reward from the LATR-generated diverse rollout group. Training exclusively with LATR throughout the entire process may thus bias the policy toward over-exploration patterns that do not generalize. To bridge this gap, we adopt a hybrid sampling strategy during RL training. At each training step, we allocate a fraction $\eta$ of rollouts to LATR and the remainder to standard Stochastic Sampling:

$$k_{\text{LATR}} = \lfloor \eta k \rceil, \quad k_{\text{std}} = k - k_{\text{LATR}}, \quad (9)$$

where $\lfloor \cdot \rceil$ denotes rounding to the nearest integer. We anneal $\eta$ exponentially over training step $i$:

$$\eta = \eta_0 \cdot \gamma^i, \quad (10)$$

with decay rate $\gamma < 1$. This ensures early-stage exploration benefits from LATR's diversity, while later stages increasingly mimic test-time behavior to reduce train-test discrepancy.

## 4 EXPERIMENTS

### 4.1 EXPERIMENTAL SETUP

To rigorously assess LATR's performance in reasoning-intensive environments, we evaluate it on two canonical domains suited for RLVR: logical reasoning and mathematical problem solving.

Table 1: Performance comparison of test correctness and average completion length on the **Countdown** dataset. ↑ indicates higher is better, while ↓ indicates lower is better. Relative improvement of LATR to Stochastic Sampling with the same policy update algorithm is marked in the parentheses, where green indicates positive improvements and red otherwise. Best results are in bold.

| Method | Correctness (%) ↑ | | Average Length ↓ | |
|---|---|---|---|---|
| | Pass@1 | Pass@8 | Pass@1 | Pass@8 |
| Qwen2.5-3B | 1.1 | 5.5 | 543 | 975 |
| + GRPO w Stochastic | 65.9 | 73.9 | 473 | 610 |
| + DAPO w Stochastic | 70.7 | 78.0 | 483 | 630 |
| + GRPO w LATR | 70.9 (+5.0) | 77.4 (+3.5) | 378 (-20%) | 469 (-23%) |
| + DAPO w LATR | **74.7** (+4.0) | **81.5** (+3.5) | **367** (-24%) | **453** (-28%) |

Table 2: Performance comparison on **DAPO Math** and **AMC 2023** dataset.

| Method | DAPO-Math (val) | | | | AMC-2023 | | | |
|---|---|---|---|---|---|---|---|---|
| | Correctness (%) ↑ | | Average Length ↓ | | Correctness (%) ↑ | | Average Length ↓ | |
| | Pass@1 | Pass@8 | Pass@1 | Pass@8 | Pass@1 | Pass@8 | Pass@1 | Pass@8 |
| Qwen2.5-3B | 5.6 | 20.1 | 938 | 2203 | 5.9 | 20.7 | 963 | 2255 |
| + GRPO w Stoch. | 24.1 | 51.3 | 880 | 1732 | 32.8 | 59.7 | **833** | 1622 |
| + DAPO w Stoch. | 26.8 | 53.1 | 1024 | 2022 | 37.8 | 62.7 | 1075 | 2116 |
| + GRPO w LATR | 28.4 (+4.3) | 51.9 (+0.6) | **853** (-3%) | **1556** (-10%) | 35.6 (+2.8) | 60.3 (+0.6) | 838 (+1%) | **1537** (-5%) |
| + DAPO w LATR | **32.5** (+5.7) | **54.1** (+2.8) | 896 (-13%) | 1880 (-7%) | **45.3** (+7.5) | **65.0** (+2.3) | 883 (-18%) | 1920 (-9%) |

**Logical Reasoning.** We adopt the Countdown dataset for both training and evaluation. Following prior work (Pan et al., 2025), we use reward $R = 0.1 \cdot R_{\text{format}} + 0.9 \cdot R_{\text{correctness}}$, where $R_{\text{format}}$ encourages outputs with proper form and $R_{\text{correctness}}$ assigns full reward for logically correct solutions.

**Mathematical Problem Solving.** Models are trained on the DAPO-Math dataset and evaluated on three additional benchmarks: MATH-500 (Hendrycks et al., 2021), AMC-2023 (MAA, 2023), and Olympiad-Bench (He et al., 2024). Consistent with Yu et al. (2025), the reward is binary: $R = 1.0$ for correct final answers, and $0$ otherwise.

**Evaluation Protocol.** For each test instance, we sample 8 independent completions. We report Pass@1 and Pass@8 correctness scores along with the average completion length to assess solution conciseness. All implementation details, including dataset descriptions, hyperparameters, and environment configurations, are provided in Appendix B.

## 4.2 TERMINATING PERFORMANCE

We provide a comprehensive comparison between LATR and Stochastic Sampling in Table 1, 2 and 3 for Countdown and Math tasks, reporting their performance and completion length on test datasets after the complete 500 steps of training. Observations are summarized as follows:

**LATR delivers consistent gains in correctness across various benchmarks on both GRPO and DAPO.** Across all task-policy combinations, LATR outperforms Stochastic Sampling in final Pass@1 scores. On the Countdown dataset, LATR improves accuracy by an average of 4.5% under both GRPO and DAPO. On the Math dataset, gains are averagely 3.8%. Notably, GRPO + LATR achieves comparable or even higher performance than DAPO + Stochastic Sampling despite DAPO's computationally intensive mechanisms such as group filtering. Moreover, DAPO + LATR achieves state-of-the-art performance on both benchmarks, reinforcing that trajectory diversity during rollout is the primary driver of performance gains. This finding aligns with ablation studies in DAPO (Yu et al., 2025), which identified rollout group filtering as the most effective component of their framework.

**LATR consistently reduces inference cost while enhancing performance.** Beyond accuracy, LATR significantly reduces the average length of generated reasoning trajectories at test time. On Countdown, completion length decreases by 22% under both GRPO and DAPO; on math datasets, we observe a 8.3% reduction. We attribute this dual benefit to LATR's core mechanism: by encouraging exploration of diverse reasoning paths during training, it exposes the policy to a broader distribution of solutions, guiding the model to internalize efficient reasoning strategies. In contrast, Stochastic Sampling tends to traverse the reasoning space sequentially within independent trajectories due to its insufficient parallel exploration, often resulting in verbose, redundant, or over-elaborated chains.

Table 3: Performance comparison on **MATH-500** and **Olympiad-Bench** dataset.

| Method | MATH-500 | | | | Olympiad-Bench | | | |
| | Correctness (%) ↑ | | Average Length ↓ | | Correctness (%) ↑ | | Average Length ↓ | |
| | Pass@1 | Pass@8 | Pass@1 | Pass@8 | Pass@1 | Pass@8 | Pass@1 | Pass@8 |
| --- | --- | --- | --- | --- | --- | --- | --- | --- |
| Qwen2.5-3B | 24.7 | 54.4 | 748 | 1690 | 8.5 | 25.8 | 1088 | 2413 |
| + GRPO w Stoch. | 58.4 | 76.7 | 657 | 1207 | 27.2 | 47.4 | 1058 | 2014 |
| + DAPO w Stoch. | 60.4 | **79.2** | 700 | 1283 | 28.1 | 47.0 | 1162 | 2193 |
| + GRPO w LATR | 61.9 (+3.5) | 77.5 (+0.8) | **594** (-10%) | **952** (-25%) | 29.5 (+2.3) | **48.2** (+0.8) | **954** (-10%) | **1728** (-14%) |
| + DAPO w LATR | **62.6** (+2.2) | 79.0 (-0.2) | 653 (-7%) | 1217 (-5%) | **30.4** (+2.3) | 47.8 (+0.8) | 1105 (-5%) | 2354 (+7%) |

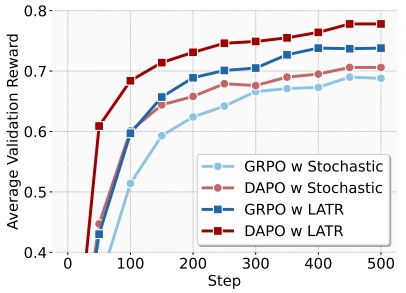 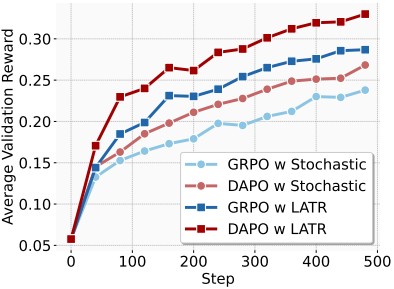

Figure 3: Learning curve comparison on Countdown (left) and DAPO-Math (right) datasets.

## 4.3 TRAINING DYNAMICS

To further investigate the RL training process with LATR and Stochastic Sampling, we analyze training dynamics by plotting validation accuracy against training step in Figure 3. The results reveal that LATR not only converges to a better solution, but does so considerably faster.

Under DAPO, Stochastic Sampling requires 450 steps to reach peak performance on the Countdown task, whereas LATR achieves the same level of accuracy by step 150, resulting in a 3× acceleration in training efficiency. On the math task, compared to step 500 for Stochastic Sampling, DAPO + LATR reaches same performance at step 240, yielding a 2× speedup. Crucially, the acceleration provided by LATR exceeds that gained by upgrading from GRPO to DAPO, despite DAPO's heavier data requirements and computational overhead per step. This suggests that LATR's enhanced exploration of diverse trajectories is able to translate into more informative policy updates per training iteration. In effect, LATR increases the sample efficiency of the RL process, enabling faster learning without architectural changes or additional data.

## 5 DISCUSSIONS

To evaluate the behavior and advantages of LATR under varying conditions, we conduct a comprehensive set of controlled experiments. Unless otherwise specified, all analyses in this section are performed using the DAPO algorithm on the Countdown dataset, with all other hyperparameters and architectural settings held consistent with the main experiments. Comparison with other rollout strategies, impact of different similarity metrics for pruning, impact of branching and pruning thresholds, analyses on efficiency, and key statistics of LATR are provided in Appendix C.

## 5.1 DIVERSITY COMPARISON

To empirically validate that LATR promotes greater diversity among reasoning trajectories within each rollout group, we conduct a comparative analysis between LATR and the baseline method Stochastic Sampling. We evaluate three variants of the Qwen2.5-3B architecture: Qwen2.5-3B, Qwen2.5-3B-Instruct, and Qwen2.5-3B trained with GRPO + LATR, which we name it Qwen2.5-LATR. This progression, from a raw pretrained model to an instruction-tuned variant and finally to a policy-optimized model incorporating LATR, enables a nuanced assessment of how LATR influences diversity across different stages of model development. In addition to standard performance metrics (Pass@1 and Pass@8), we also evaluate the average number of distinct final answers per rollout group. Two answer expressions of Countdown are considered distinct if their evaluated numerical

outcomes differ. This ensures that diversity is measured in terms of semantic rather than syntactic variation.

As shown in Table 4, LATR consistently yields higher Pass@8 scores and a greater number of distinct answers per rollout group across all three model variants compared to Stochastic Sampling. These results support our claim that LATR enhances intra-group trajectory diversity, thereby facilitating more effective policy learning through broader exploration of the solution space.

Table 4: Diversity comparison between Stochastic Sampling and LATR.

| Method | Pass@1 | Pass@8 | # Ans. |
|---|---|---|---|
| Qwen2.5-3B + Stoch. | 5.8 | 28.9 | 6.3 |
| Qwen2.5-3B + LATR | 6.1 | 30.7 | 6.9 |
| Qwen2.5-3B-Instruct + Stoch. | 9.4 | 35.2 | 6.4 |
| Qwen2.5-3B-Instruct + LATR | 10.9 | 40.6 | 6.9 |
| Qwen2.5-3B-LATR + Stoch. | 70.9 | 77.4 | 2.6 |
| Qwen2.5-3B-LATR + LATR | 68.9 | 79.9 | 3.0 |

## 5.2 EFFECT OF DIFFERENT COMPONENTS

We dissect the contributions of LATR's core components through an ablation study. Specifically, we evaluate four variants of LATR: (1) random branching, (2) random pruning, (3) no pruning, and (4) token-level lookahead in place of trajectory-level lookahead. In the random variants, the branching or pruning ratio is matched to the average ratio observed in the full LATR throughout training. In the token-level lookahead variant, pruning decisions are made solely based on

Table 5: Performance comparison on Stochastic Sampling and variants of LATR.

| Method | Pass@1 | Pass@8 |
|---|---|---|
| Stochastic | 70.7 | 78.0 |
| LATR w rand branch | 69.6 | 75.8 |
| LATR w rand prune | 72.5 | 79.2 |
| LATR w/o prune | 71.0 | 78.7 |
| LATR w token-level lookahead | 72.1 | 80.4 |
| LATR | **74.7** | **81.5** |

the next token: a branch is pruned if the next tokens across trajectories are identical. This design enables us to isolate the effects of structured branching and similarity-based pruning on overall performance. Our findings are summarized below.

**Random branching leads to unstable training and degrades final performance.** As shown in Table 5, LATR with random branching performs even worse than Stochastic Sampling. We observe that the KL divergence between the policy model and the base (reference) policy rises to as high as 1.0 within just 50 training steps, signaling severe off-policy behavior. This instability stems from uncontrolled branching. Without the probability thresholds imposed by our method, the model may generate extremely low-probability sequences that diverge significantly from the base policy, thereby disrupting the learning process.

**Both random and no pruning yield suboptimal results.** The variants of LATR without pruning and with random pruning achieve only modest improvements over Stochastic Sampling, confirming that branching alone enhances exploration by diversifying rollout trajectories. However, the full LATR outperforms both. This performance gap is primarily attributable to trajectory-level redundancy, as rollout groups generated without pruning or with random pruning frequently contain sequences that follow similar reasoning paths, reducing effective diversity and leading to inefficient policy updates. Moreover, the comparison between the no-pruning and random-pruning variants highlights budget exhaustion as another critical factor. Without pruning, the fixed rollout budget $k$ is quickly depleted in early generation steps, leaving insufficient capacity for exploration in later stages.

**Token-level lookahead underperforms trajectory-level lookahead.** Although token-level lookahead outperforms both Stochastic Sampling and the no-pruning variant, it falls significantly short of the full LATR model. This deficit stems from its limited ability to capture trajectory divergence. Pruning decisions based solely on the next token are often inaccurate, leading to the premature removal of potentially valuable branches and degrading rollout quality.

In summary, while branching provides a robust mechanism for exploration, dynamic, similarity-aware pruning serves as a crucial regulator: it ensures that the exploration budget is allocated meaningfully across the generation process and effectively mitigates redundant trajectories.

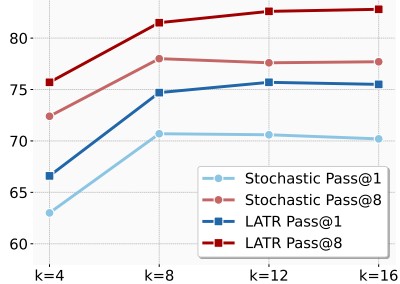 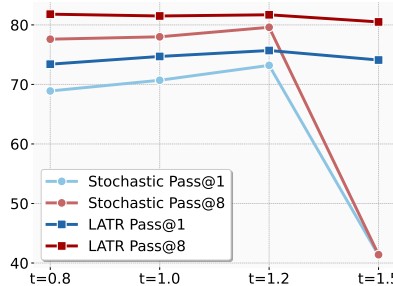

Figure 4: Comparison of test correctness with different rollout number $k$ and temperature $t$ (%).

## 5.3 SCALABILITY WITH ROLLOUT NUMBER

The rollout budget $k$ fundamentally constrains the scope of exploration in RLVR training. We evaluate how LATR and Stochastic Sampling scale with increasing $k \in \{4, 8, 12, 16\}$. Results in Figure 4 reveal two critical trends:

1) LATR consistently outperforms Stochastic Sampling at every value of $k$, demonstrating robustness to budget constraints.
2) While Stochastic Sampling performance plateaus at $k = 8$, LATR continues to improve up to $k = 12$, indicating a higher effective capacity for leveraging additional rollouts.

This suggests that LATR not only uses its budget more efficiently but also raises the performance ceiling of the system, enabling gains from larger $k$ values that Stochastic Sampling cannot exploit.

## 5.4 IMPACT OF DIFFERENT SAMPLING TEMPERATURES

In standard RL frameworks with Stochastic Sampling, the sampling temperature $t$ governs the exploration-exploitation trade-off: higher $t$ increases stochasticity and thus exploration, but risks degrading rollout quality. In contrast, LATR delegates exploration primarily to its branching-and-pruning mechanism, using $t$ only to modulate stochastic fallback and hybrid rollouts, which is described in Section 3.3.

We evaluate performance across $t \in \{0.8, 1.0, 1.2, 1.5\}$. As shown in Figure 4, both methods peak near $t = 1.2$, suggesting this is optimal for the base policy. Notably, LATR achieves superior performance at every $t$, and exhibits lower variance across temperatures.

This robustness stems from LATR's architectural decoupling: exploration is driven by structural diversity (branching + pruning), not sampling noise. Consequently, LATR is less sensitive to suboptimal temperature tuning.

## 5.5 GENERALIZABILITY ACROSS DIFFERENT BASE MODELS

To evaluate the generalizability of LATR across diverse base models, we conduct additional experiments using additional models from different series and scales, specifically Qwen2.5-7B and Qwen3-1.7B-Base. As shown in Table 6, LATR consistently outperforms Stochastic Sampling across all evaluated models, demonstrating the broad applicability and robustness of our proposed method.

Table 6: Performance comparison of LATR and Stochastic Sampling on different base models.

| Method | Correctness (%) ↑ | | Average Length ↓ | |
| --- | --- | --- | --- | --- |
| | Pass@1 | Pass@8 | Pass@1 | Pass@8 |
| Qwen2.5-7B | 1.1 | 5.4 | 556 | 997 |
| Qwen2.5-7B + Stochastic | 70.9 | 79.8 | 522 | 722 |
| Qwen2.5-7B + LATR | **76.0** | **82.1** | **396** | **508** |
| Qwen3-1.7B-Base | 2.0 | 9.5 | 542 | 916 |
| Qwen3-1.7B-Base + Stochastic | 66.0 | 75.5 | 521 | 673 |
| Qwen3-1.7B-Base + LATR | **67.8** | **77.6** | **494** | **662** |

# 6 RELATED WORK

## 6.1 REINFORCEMENT LEARNING WITH VERIFIABLE REWARDS

Reinforcement Learning with Verifiable Rewards (RLVR) has emerged as a powerful alternative for tasks with verifiable results (Lambert et al., 2024). In RLVR, the reward signal is derived from an external verifier, providing an objective measure of a trajectory's success. Within RLVR, GRPO (Shao et al., 2024) have become the state-of-the-art solution. Rather than relying on a learned value model, it compares trajectories within a sampling group and updates policy based on relative success.

Following this line of research, many seek to improve the performance of GRPO. DAPO introduces clip higher technique and removes RL constraints to enable aggressive policy updates towards correct reasoning. GSPO (Zheng et al., 2025) proposes sequence-level rewards to smooth and stabilize learning. These innovations on policy update are orthogonal to the rollout strategy, so LATR is fully compatible with these methods. Replacing vanilla rollouts with LATR-generated trajectories yields additive performance improvements, as we demonstrate empirically.

A few works have also touched upon the rollout strategy, though typically as a secondary component. DAPO (Yu et al., 2025) proposes a group filtering strategy to oversample and discard groups with identical rewards. ProRL (Liu et al., 2025) increases the sampling temperature to obtain more diverse rollout sequences. Despite these advancements, these methods only address trajectory-level in-group diversity indirectly. Their reliance on token-level stochastic sampling is prone to generating semantically redundant reasoning paths, a limitation our work directly confronts.

More recently, two contemporaneous works integrate tree search into RLVR. TreeRL (Hou et al., 2025) and TreePO (Li et al., 2025) propagate sparse binary outcome rewards backward through the reasoning tree to derive dense process rewards that guide policy updates. TreePO additionally enhances generation efficiency by reusing shared prefixes and pruning unpromising branches early in the rollout process. While both methods leverage tree-based structures, their objectives differ fundamentally from ours, as they primarily aim to refine reward estimation or improve computational efficiency. In contrast, we adopt tree-search to explicitly foster and compare diverse reasoning trajectories within a rollout group. This trajectory-level diversity enriches the reward signal by capturing a broader spectrum of potential outcomes, thereby enhancing policy learning.

## 6.2 LOOKAHEAD REASONING FOR LLMS

Recent work has increasingly explored lookahead-based reasoning strategies in LLMs, with their majorly focus on inference-time approaches or offline data construction. Tree-of-Thoughts (ToT) (Yao et al., 2023) pioneered this direction by generating multiple reasoning branches at each step and selecting the most promising path using an external reward model. Subsequent methods such as MCTS-DPO (Xie et al., 2024) and ReST-MCTS (Zhang et al., 2024) extend this idea by integrating Monte Carlo tree search with lookahead estimation to decompose sparse, instance-level rewards into dense, step-level supervision signals. Quiet-STaR (Zelikman et al., 2024) also leverages a lookahead mechanism, generating token-wise rationales that anticipate future text and optimizing them based on their contribution to correct continuations.

While these works share the common ingredient of lookahead search, their objectives differ fundamentally from ours. Our primary goal in employing lookahead tree search is not reward propagation or step-level supervision, but rather to explicitly compare and promote trajectory-level diversity among rollouts for the same problem. This explicit focus on trajectory-level diversity distinguishes our method from prior lookahead approaches in LLMs.

# 7 CONCLUSION

In this work, we present Lookahead Tree-Based Rollout, a novel rollout strategy that explicitly promotes trajectory-level diversity in RLVR by dynamically branching at high-uncertainty tokens and pruning non-divergent paths via lookahead simulation. By moving beyond token-level sampling heuristics, LATR enriches policy learning signals, accelerating training convergence while improving final performance by a large margin across different benchmarks. Our work demonstrates that trajectory-level rollout diversity is key to scaling RLVR effectively and efficiently.

## REPRODUCIBILITY STATEMENT

To better support reproducibility, we explain all the details to reproduce our results in Section B.3, including the parameters for our methods, training details, environment and framework versions.

## ACKNOWLEDGMENTS

This research is supported in part by the Office of the Director of National Intelligence (ODNI), Intelligence Advanced Research Projects Activity (IARPA), via the HIATUS Program contract #2022-22072200006, the Defense Advanced Research Projects Agency with award HR00112220046, and NSF IIS 2048211. We would like to thank all the collaborators in USC INK research lab for their constructive feedback on the work.

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

## A    LLM USAGE

In the course of preparing this manuscript and supporting materials, we leveraged large language models (LLMs) as auxiliary tools to enhance the efficiency and quality of non-core research tasks. Specifically, LLMs were employed in two primary capacities:

1) **Language polishing**: We used LLMs to assist in proofreading, grammatical correction, and stylistic refinement of the manuscript's prose.
2) **Boilerplate and utility code generation**: For ancillary implementation tasks, such as file I/O wrappers, format converters, or logging utilities, we used LLMs to accelerate prototyping.

## B    DETAILS ON EXPERIMENT SETUP

In this section, we detail the datasets, evaluation protocols, and implementation configurations.

### B.1    DATASETS AND TASK FORMULATIONS

**Logical Reasoning.**  We adopt the Countdown dataset (Pan et al., 2025), which challenges models to construct arithmetic expressions from a given set of integers that evaluate exactly to a target number. Following Pan et al. (2025), we define a two-component reward function:

$$R = 0.1 \cdot \mathbb{I}_{\text{format}} + 0.9 \cdot \mathbb{I}_{\text{correct}}, \tag{11}$$

where $\mathbb{I}_{\text{format}}$ indicates syntactic validity and $\mathbb{I}_{\text{correct}}$ indicates semantic correctness. Models are trained on the training split and evaluated on the official test set.

**Mathematical Problem Solving.**  For mathematical reasoning, we train on the DAPO Math dataset (Yu et al., 2025), a curated collection of problems drawn from diverse sources. Consistent with Yu et al. (2025), the reward is binary:

$$R = \mathbb{I}_{\text{correct}}, \tag{12}$$

awarding 1.0 only for exact numerical matches.

To ensure broad generalization, we evaluate not only on DAPO Math's held-out validation set, which is manually partitioned with 1,024 samples, but also on three established external benchmarks, including MATH500 (Hendrycks et al., 2021), AMC2023 (MAA, 2023), and OlympiadBench (He et al., 2024).

To maintain consistency across datasets with heterogeneous answer formats, following Yu et al. (2025), we apply a standardized answer normalization pipeline that maps all results to integers. We construct a comprehensive few-shot prompt that instructs Gemini-2.5-pro (Team et al., 2025) to apply a set of deterministic heuristics according the original answer's format. These heuristics include: (1) for structured non-integer answers like fractions $(p/q)$ or radicals $(k + m\sqrt{n})$, rephrasing the question to ask for the sum of their components (e.g., $p + q$ or $k + m + n$); (2) for symbolic expressions, either summing the coefficients of simple polynomials or evaluating complex functions when assigning the variables (e.g., $x = 2$); and (3) for multi-part or multiple-choice answers, asking for the sum of solutions or the 0-indexed position of the correct choice. The few shot prompt applied is provided in Figure 6.

### B.2    EVALUATION PROTOCOL

We perform stochastic sampling on the trained model for a fixed 8 times for each sample in the evaluation datasets, and report Pass@1 (the average accuracy over a single sampled completion per

question) and Pass@8 (the accuracy of the best solution among 8 independently sampled completions per question). In addition to correctness, we also include average completion length for both Pass@1 and Pass@8 to quantify test-time computational cost and efficiency.

### B.3 Implementation Details

**Sampling and Rollout Parameters.** We set our sampling parameters following Yu et al. (2025); Pan et al. (2025). During training, we sample rollouts with temperature = 1.0, top-$k$ = $-1$, and top-$p$ = 1.0 to encourage exploration. During evaluation, we use temperature = 0.6, top-$k$ = 20, and top-$p$ = 0.95 for calibrated diversity. Each training step involves $k = 8$ rollouts per prompt. Maximum completion lengths are set to 1,024 tokens for Countdown and 8,192 tokens for math problems.

**Algorithmic Parameters for LATR.** Hybrid rollout coefficient $\eta_0 = 1.0$, decaying per-step via $\eta_t = \eta_0 \cdot \gamma^t$, with $\gamma = 0.985$ (Countdown) and $\gamma = 0.995$ (Math). For branching thresholds, absolute probability threshold $\tau_{\text{abs}} = 0.25$, relative probability threshold $\tau_{\text{rel}} = 0.15$, and edit-distance threshold $\tau_{\text{ed}} = 0.4$. Lookback step $r$ is $\{20, 30, 50\}$, which means we enforce conditions on all of the 3 lookback windows, and all should be satisfied for a branch to be kept.

**Training Hyperparameters.** For training parameters, global data batch size is 256, global mini batch size is 256, local micro batch size is 8 for Countdown and 4 for DAPO Math, clip ratio is 0.2, KL penalty $\beta$ is 0.01. For DAPO, clip ratio high is 0.28, low is 0.2, and oversampled data generation batch size is 384. We train the Qwen2.5-3B base model on both datasets for a fixed 500 steps with AdamW optimizer and a constant learning rate of 1e-6. All our experiments are performed with VeRL-0.5.0 framework (Sheng et al., 2024) on 8× NVIDIA H200 GPUs with mixed precision.

## C Additional Analyses

### C.1 Comparison with Alternative Rollout Strategies

To further demonstrate the effectiveness of LATR, we compare it against two other baseline rollout strategies:

1) **Rollout Down-sampling (RDS)** (Xu et al., 2025): Similar to the group filtering in DAPO, RDS also seeks to enhance trajectory diversity in a post-hoc manner. Specifically, it first generates $k = 16$ trajectories via standard rollout and then selects the 8 most diverse trajectories for policy updates. The selection is implemented greedy, using the average of sentence-level BLEU and ROUGE scores as trajectory similarity measures.

2) **Entropy Guided Tree Search (EPTree)** (Hou et al., 2025): Proposed in TreeRL, EPTree constructs a rollout tree to support fine-grained reward estimation and optimization during policy updates. After generating $M$ complete sequences, it identifies the top-$N$ tokens with the highest entropy and re-generates continuations $T$ times from each of these tokens, yielding a total of $M \times (N \times T + 1)$ sequences. Following the setup in TreeRL, we use $(M, N, T) = (4, 2, 1)$, resulting in 10 sequences per rollout. To ensure a fair comparison, we randomly sample 8 trajectories from these 10 for policy updates. We directly use the official code from TreeRL and integrates EPTree rollout into the VeRL training framework.

As shown in Table 7, LATR consistently outperforms both RDS and EP-Tree across both GRPO and DAPO policy update algorithms. Notably, while GRPO combined with RDS yields improvements over Stochastic Sampling due to enhanced trajectory diversity, the same combination under DAPO fails to surpass Stochastic Sampling's performance. Further analysis reveals that DAPO + RDS leads to un-stable training dynamics, marked by performance degradation and sharp increases in KL divergence during later

Table 7: Performance comparison of different rollout strategies on the Countdown dataset.

| Method | Correctness (%) ↑ | | Average Length ↓ | |
|---|---|---|---|---|
| | Pass@1 | Pass@8 | Pass@1 | Pass@8 |
| GRPO w Stoch. | 65.9 | 73.9 | 473 | 610 |
| GRPO w RDS | 68.7 | 75.7 | **365** | 489 |
| GRPO w EPTree | 65.3 | 73.5 | 471 | 599 |
| GRPO w LATR | **70.9** | **77.4** | 378 | **469** |
| DAPO w Stoch. | 70.7 | 78.0 | 483 | 630 |
| DAPO w RDS | 68.5 | 74.4 | **348** | 462 |
| DAPO w EPTree | 66.3 | 74.6 | 450 | 595 |
| DAPO w LATR | **74.7** | **81.5** | 367 | **453** |

training stages. This instability likely
stems from the diversity-oriented se-
lection mechanism, which biases towards selecting low-probability trajectories, thereby increasing
off-policy risk. When combined with DAPO, which already promotes diversity through group-based
filtering, this effect is amplified, ultimately contributing to model collapse.

In contrast, the underwhelming performance of EPTree suggests that the gains reported in TreeRL
primarily arise from its novel policy update mechanism rather than its rollout strategy. Specifically,
TreeRL employs Monte Carlo Tree Search (MCTS) to estimate fine-grained rewards for individual
tree branches by propagating sparse binary outcome rewards backward through the tree, enabling
targeted optimization of intermediate reasoning steps. By contrast, LATR improves RL performance
by enhancing trajectory-level diversity without requiring modifications to the underlying policy
update algorithm.

## C.2 IMPACT OF DIFFERENT SIMILARITY METRICS

As described in Equation 7, the main experiments employ edit distance as the similarity measure to
identify and prune redundant trajectory branches. In principle, however, numerous alternative metrics
could be used to assess the divergence between partial sequences. To investigate this, we evaluate
three additional similarity measures in this section:

1) **ROUGE-L**: defined as the ratio of the length of the longest common subsequence to the sequence
   length.
2) **Suffix matching**: defined as the ratio of the length of the longest suffix of one sequence that
   appears anywhere in the other sequence to the sequence length.
3) **Embedding-based**: the cosine similarity of the sequence embeddings calculated by the model
   Qwen3-Embedding-0.6B.

For each metric, we conduct experiments while tuning the pruning threshold to identify its optimal
value. The best results, summarized in Table 9, show that embedding-based similarity yields the
weakest performance, while all other token-level metrics achieve comparable and better final accuracy.
The failure of embedding-based similarity is likely to stem from the inability for embedding models to
capture fine-grained logical distinctions, since these models are usually trained to discern topic-level
differences. Therefore, Given its simplicity and competitive efficacy, we retain edit distance as our
pruning criterion.

## C.3 IMPACT OF BRANCHING AND PRUNING THRESHOLDS

To investigate the impact of the pruning threshold
$\tau_{ed}$, we conduct a series of experiments with $\tau_{ed} \in$
$\{0.3, 0.4, 0.5, 0.6\}$. The results are summarized in Table
8. We observe that $\tau_{ed} = 0.4$ yields the best performance,
while both lower and higher values lead to a decline in
results. This behavior can be attributed to the trade-off
imposed by the pruning threshold: excessively high values
result in overly similar trajectories, which consume the
rollout budget without promoting exploration, whereas
excessively low values impose overly stringent constraints,
leading to an insufficient number of viable trajectories.

For branching thresholds $\tau_{abs}, \tau_{rel}$, we experiment by fix-
ing one threshold while adjusting the other one. As shown
in Table 8, $(\tau_{abs}, \tau_{rel}) = (0.25, 0.15)$ yields the best per-
formance, corroborating the suitability of our selected
hyperparameters.

Table 8: Performance comparison of
different thresholds.

| Threshold | Pass@1 | Pass@8 |
|---|---|---|
| $\tau_{abs} = 0.2$ | 71.4 | 77.2 |
| $\tau_{abs} = 0.25$ | 74.7 | 81.5 |
| $\tau_{abs} = 0.3$ | 72.3 | 78.5 |
| $\tau_{rel} = 0.1$ | 72.0 | 77.7 |
| $\tau_{rel} = 0.15$ | 74.7 | 81.5 |
| $\tau_{rel} = 0.2$ | 72.1 | 77.6 |
| $\tau_{ed} = 0.3$ | 73.8 | 80.5 |
| $\tau_{ed} = 0.4$ | 74.7 | 81.5 |
| $\tau_{ed} = 0.5$ | 74.1 | 81.4 |
| $\tau_{ed} = 0.6$ | 73.4 | 79.8 |

## C.4 EFFICIENCY ANALYSIS

The search tree in LATR is bounded by a maximum width corresponding to the rollout number
$k$. Unlike Stochastic Sampling, which perform forward passes on each sequence independently at

Table 9: Performance comparison of different similarity metrics.

| Method | Correctness (%) ↑ | | Average Length ↓ | |
|---|---|---|---|---|
| | Pass@1 | Pass@8 | Pass@1 | Pass@8 |
| Edit Distance | 74.7 | 81.5 | 367 | 453 |
| ROUGE-L | 73.9 | 80.5 | 390 | 486 |
| Suffix Matching | 74.9 | 81.7 | 388 | 493 |
| Embedding-based | 72.9 | 79.8 | 369 | 445 |

each step, LATR dynamically branches and prunes sequences, resulting in a sparser tree structure, particularly during early generation stages. Consequently, the actual number of FLOPs consumed by LATR is strictly less than that of Stochastic Sampling for the same settings.

Empirically, LATR exhibits a modest slowdown in generation speed during RL training compared to Stochastic Sampling, as shown in Figure 5. Specifically, LATR runs approximately 10% slower per step than Stochastic Sampling with the same configuration. However, compared with DAPO with Stochastic Sampling, GRPO with LATR is able to achieve comparable performance in shorter training time. Additionally, considering the averagely $2.3\times$ training speedup (as introduced in Section 4.3), LATR is able to achieve higher performance in less total training time. This suggests that the algorithmic gains of LATR outweigh its runtime penalties in end-to-end training scenarios.

Further profiling reveals that the runtime overhead in LATR primarily stems from the sequential computation patterns during the branching and pruning phase. Unlike Stochastic Sampling, which processes contiguous batched inputs, LATR dynamically inserts and removes sequences during tree expansion and pruning, so indexing and comparisons are performed per sequence rather than in a fully batched manner. Targeted optimizations, analogous to PagedAttention (Kwon et al., 2023) for Stochastic Sampling, are likely to mitigate the overhead. While promising, such engineering improvements lie outside the scope of this work and are left to future efforts.

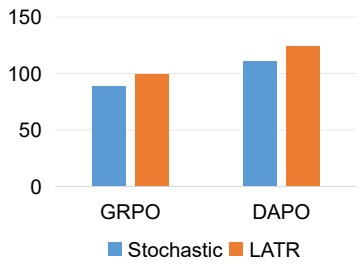

Figure 5: Comparison of average consumed time per training step under different settings (second).

## C.5 ADDITIONAL STATISTICS FOR LATR

To further elucidate the behavior of our proposed method, we report two key statistics: the average branching ratio, which is the proportion of tokens at which new reasoning branches are initiated relative to the total number of generated tokens, and the average satu-

Table 10: Key statistics for LATR.

| Model | Branching Ratio | Saturation Length |
|---|---|---|
| Qwen2.5-3B | 0.101 | 65 |
| Qwen2.5-3B-Instruct | 0.039 | 102 |
| Qwen2.5-3B-LATR | 0.044 | 132 |

ration length, defined as the number of tokens generated before early stopping is triggered. Following the setup in Section 5.1, we present these metrics for the same three model variants: Qwen2.5-3B, Qwen2.5-3B-Instruct, and Qwen2.5-LATR, enabling a consistent and comprehensive analysis across model stages.

As shown in Table 10, the branching ratios are consistently low across all models, indicating conservative branching behavior. Moreover, the average saturation length is notably shorter than the maximum completion length of 1,024 tokens. This observation aligns with prior findings (Shao et al., 2025), which suggest that the initial segments of a reasoning chain are often most critical in determining the final outcome.

You are an expert AI assistant specializing in mathematics. You must strictly return the result in a valid JSON format.
IMPORTANT: In all JSON strings, every literal backslash `\` for LaTeX commands must be escaped as `\\`. For example, `\frac` must be written as `\\frac`.

--- Task ---
Given a math problem with its non-integer answer, rephrase it into a new problem whose solution is a single integer.

--- Guidelines ---
1.  **For non-integer numerical answers**, convert them to a single integer in base 10 using the following rules:
    * **Fractions:** For a simplified fraction $\\frac{a}{b}$, ask for the value of a + b. For a mixed number $a\\frac{b}{c}$, ask for a + b + c.
    * **Radicals:** For an answer in the form $k + m\\sqrt{n}$, ask for the value of k + m + n. If k=0, ask for m + n.
    * **Complex Numbers:** For a complex number a + bi, ask for the value of a + b.
    * **Coordinate Points:** For an ordered pair $\\left( a, b\\right)$, ask for the value of a + b.
    * **Multiples of Pi:** For an answer k * pi, ask for the value of k.
    * **Other Number Bases:** For an answer d in base b (e.g., 52_8), ask for its value in base 10.
2.  **For Symbolic Expressions and Functions**, choose the appropriate strategy based on the expression's type:
    * **(Strategy A) Component Analysis for Simple Polynomials and Fixed Structures:**
        For answers that are simple linear combinations, polynomials, or have a fixed symbolic structure, prompt for the sum of their key integer components. **This should be prioritized for simple expressions.**
        * **Applies to:** $a \cdot p + b \cdot q$, $n^2 + an + b$
        * **And fixed forms like:** $a^b$, $a!$, `$a! / b^c$
        * **Example Action:** For $2n + 3$, ask for the sum of coefficients and constants (`2+3`).
    * **(Strategy B) Evaluation for Complex or Non-Polynomial Functions:**
        For answers involving complex functions (e.g., binomials, floors) or defined as pairs of functions, prompt for the expression's numerical value at small, specified integer inputs.
        * **Applies to:** $\\binom{2n}{n}$, $mn - \\lfloor m/2 \\rfloor$, or function pairs like $f(n), g(n)$.
        * **Example Action:** For $\\binom{2n}{n}$, ask for its value at $n=2$.
3.  **For multiple-choice or named answers**:
    * Convert the list of possible choices into a 0-indexed list and ask for the index of the correct answer.

--- Requirements ---
1. Please modify the instruction of the output format if necessary. Except that, do not change the content of the original problem.
2. Adhere to the JSON format:
{"question": "The complete rewritten question text", "answer": The integer answer}

--- Examples ---
{{Examples}}

--- Now, please process the following problem ---
Original Problem:

Figure 6: Prompt for data transformation.

