# OpenReview forum: "Lookahead Tree-Based Rollouts for Enhanced Trajectory-Level Exploration in Reinforcement Learning with Verifiable Rewards"
_ICLR.cc/2026/Conference — ICLR 2026 Poster_

### Official Review · Reviewer_Vq12 · 2025-10-26

**Soundness:** 2
**Presentation:** 3
**Contribution:** 2
**Rating:** 4
**Confidence:** 2

**Summary:**

This paper introduces Lookahead Tree-Based Rollout (LATR) to enhance trajectory-level diversity among rollouts in a group to improve the performance of GRPO and DAPO in policy refinement.

**Strengths:**

1- The paper is well written and easy to follow.
2- The diversity problem in RL is well known, and this paper clearly identifies it within the RLVR paradigm.

**Weaknesses:**

1- The literature review is not thorough. Many approaches that combine look-ahead reasoning with LLMs are not mentioned.

2- The baselines are neither strong nor state-of-the-art. Although the paper cites works addressing diversity at the token level, the comparisons use only stochastic sampling; it is unsurprising that look-ahead search would outperform simple stochastic sampling.

3- The performance gains are not substantial, especially in terms of correctness.

4- Although inference cost is reduced, there is no comparison of training costs between the stochastic and look-ahead approaches. I would expect the look-ahead method to require significantly more training time.

**Questions:**

1- The paper states that the tree-search approach is inspired by MCTS. Could you clarify how your tree search is similar to the MCTS used in AlphaZero [1]?



1- Mastering the game of Go with deep neural networks and tree search.

---

> ### Author Response · Authors · 2025-11-21
> **Response to Reviewer Vq12 [1/3]**
>
> Thank you for your appreciation of our good writing and insightful motivation. For the concerns, we detail our response below. We would appreciate it if you could let us know whether your concerns are addressed.
>
>
> $~$
>
> ---
>
> $~$
>
>
> > ***Q1: The literature review is not thorough. Many approaches that combine look-ahead reasoning with LLMs are not mentioned.***
>
> **A1:** We appreciate the reviewer’s feedback and acknowledge the growing body of work on lookahead-based reasoning in LLMs. Indeed, several recent methods have explored this direction, with their majorly focus on inference-time approaches or offline data construction. Tree-of-Thoughts (ToT) [3] pioneered this direction by generating multiple reasoning branches at each step and selecting the most promising path using an external reward model. Subsequent methods such as MCTS-DPO [4] and ReST-MCTS [5] extend this idea by integrating Monte Carlo tree search with lookahead estimation to decompose sparse, instance-level rewards into dense, step-level supervision signals. Quiet-STaR [6] also leverages a lookahead mechanism, generating token-wise rationales that anticipate future text and optimizing them based on their contribution to correct continuations.
>
> While these works share the common ingredient of lookahead search, their objectives differ fundamentally from ours. Our primary goal in employing lookahead tree search is not reward propagation or step-level supervision, but rather to **explicitly compare and promote trajectory-level diversity** among rollouts for the same problem. This diversity enriches the reward signal by capturing a broader spectrum of potential outcomes, thereby improving the quality of policy learning. This explicit focus on trajectory-level diversity distinguishes our method from prior lookahead approaches in LLMs.
>
> [1] DAPO: An Open-Source LLM Reinforcement Learning System at Scale
>
> [2] ProRL: Prolonged Reinforcement Learning Expands Reasoning Boundaries in Large Language Models
>
> [3] Tree of thoughts: Deliberate problem solving with large language models.
>
> [4] Monte carlo tree search boosts reasoning via iterative preference learning.
>
> [5] Rest-mcts*: LLM self-training via process reward guided tree search.
>
> [6] Quiet-star: Language models can teach themselves to think before speaking.
>
> [7] TreeRL: LLM Reinforcement Learning with On-Policy Tree Search
>
> [8] TreePO: Bridging the Gap of Policy Optimization and Efficacy and Inference Efficiency with Heuristic Tree-based Modeling

---

> ### Author Response · Authors · 2025-11-21
> **Response to Reviewer Vq12 [2/3]**
>
> > ***Q2: The baselines are neither strong nor state-of-the-art. Although the paper cites works addressing diversity at the token level, the comparisons use only stochastic sampling; it is unsurprising that look-ahead search would outperform simple stochastic sampling.***
>
> **A2:** As outlined in Section 2, the RLVR framework comprises two orthogonal components: **rollout generation** and  **policy update**. Existing RLVR methods such as DAPO [1], GSPO [9], and ProRL [2] primarily innovate on the policy update side and generally rely on standard token-level stochastic sampling for rollouts. Although some works employ heuristics that indirectly increase trajectory diversity (e.g., higher sampling temperatures in ProRL or group filtering in DAPO), these techniques remain within the same stochastic sampling paradigm and are not central to their primary contributions.
>
>  Our work is orthogonal to these efforts. LATR serves as a **drop-in replacement** for stochastic sampling in the rollout stage, effectively improving rollout diversity (while keeping the policy update fixed) and yielding better RLVR performance. As shown in Appendix C.2, LATR consistently outperforms stochastic sampling across a wide range of temperatures. We validate this on top of two representative policy-update algorithms, GRPO and DAPO, and LATR can seamlessly be integrated with other policy update strategies such as GSPO and Pass@k Training [10]. Notably, DAPO+ LATRgenerally outperformsDAPO + stochastic sampling by 4.2% across benchmarks, even though DAPO already gains 3.7% performance improvement from policy update optimizations. This further demonstrates the general effectiveness of improving the rollout component itself.
>
> **[More Comparisons]** To further address the reviewer’s concern about stronger diversity baselines, we conduct additional experiments on the Countdown task using two explicit diversity-enhancing rollout strategies:
> (1) Rollout Down-sampling [11]: generate 16 candidate rollouts and greedily select the 8 most distinct ones;
> (2) EPTree rollout from TreeRL [7]: iteratively re-generate sequences anchored at high-entropy tokens.
>
> As summarized in Table 1, LATR achieves the best performanceamong all methods, validating the effectiveness of our method. These results and the corresponding analyses have been added to Appendix C.3 in the revised manuscript.
>
> Table 1
>
> | Method          | Pass@1 | Pass@8 | Avg.Len (Pass@1) | Avg.Len (Pass@8) |
> | ------------- | ------------------------- | ------------------------- | ------------------------ | ------------------------ |
> | GRPO w Stoch. | 65.9                      | 73.9                      | 473                      | 610                      |
> | GRPO w RDS    | 68.7                      | 75.7                      | **365**            | 489                      |
> | GRPO w EPTree | 65.3                      | 73.5                      | 471                      | 599                      |
> | GRPO w LATR   | **70.9**            | **77.4**            | 378                      | **469**            |
> | DAPO w Stoch. | 70.7                      | 78.0                      | 483                      | 630                      |
> | DAPO w RDS    | 68.5                      | 74.4                      | **348**            | 462                      |
> | DAPO w EPTree | 66.3                      | 74.6                      | 450                      | 595                      |
> | DAPO w LATR   | **74.7**            | **81.5**            | 367                      | **453**            |
>
> [9] Group Sequence Policy Optimization
>
> [10] Pass@k Training for Adaptively Balancing Exploration and Exploitation of Large Reasoning Models
>
> [11] Not All Rollouts are Useful: Down-Sampling Rollouts in LLM Reinforcement Learning

---

> ### Author Response · Authors · 2025-11-21
> **Response to Reviewer Vq12 [3/3]**
>
> > ***Q3: The performance gains are not substantial, especially in terms of correctness.***
>
> **A3:** In the context of RLVR, the improvements delivered by LATR are both meaningful and practically significant. Compared to stochastic sampling, LATR delivers (1) 131% faster training iteration, (2) 10% fewer inference tokens, and (3) an average **4.2% absolute improvement in correctness** across benchmarks. For reference, DAPO [3], a well-regarded and widely adopted RLVR framework, achieves **a 3.7% correctness gain** over GRPO at the cost of slower training and higher inference overhead. LATR achieves a larger correctness gain while simultaneously reducing inference cost and accelerating training.
>
> $~$
>
> ---
>
> $~$
>
> > ***Q4: Although inference cost is reduced, there is no comparison of training costs between the stochastic and look-ahead approaches. I would expect the look-ahead method to require significantly more training time.***
>
> **A4:** We have already provide a detailed analysis of training efficiency in Appendix C.6. From a theoretical perspective, LATR incurs **fewer FLOPs** than stochastic sampling because its sparse tree structure eliminates unnecessary branches and avoids backtracking overhead. Empirically, we do observe a modest  **~10% slowdown** , which we attribute to less cache-friendly memory access patterns introduced by branching and pruning. However, this small overhead is outweighed by LATR’s **2.3× faster convergence,** and the total training time required to achieve the same or better performance is lower for LATR overall.
>
> $~$
>
> ---
>
> $~$
>
> > ***Q5: The paper states that the tree-search approach is inspired by MCTS. Could you clarify how your tree search is similar to the MCTS used in AlphaZero?***
>
> **A5:** In standard MCTS, each iteration consists of three sequential stages: (1) Expansion, which creates child nodes according to current valid actions; (2) Simulation, which complete a random playout from the child nodes; and (3) Backpropagation, which propagate the playout outcome upward to update information of the nodes on the path from the child node to the root.
>
> Inspired by this three-stage procedure, we design our LATR as follows: (1) Branching, which creates new branches when the probability thresholds are satisfied; (2) Simulation, where new branches continue generation for a lookahead window; and (3) Pruning, where similar branches are pruned according to the simulation results.
>
> Unlike AlphaZero-style MCTS, LATR does not perform backpropagation or maintain explicit visit counts, as our goal is not value estimation but the efficient generation of diverse, high-quality rollouts for policy training. Nevertheless, the conceptual parallels in structured exploration make MCTS a natural source of inspiration.

---

> ### Author Response · Authors · 2025-11-27
> **We look forward to your response**
>
> Thanks again for your valuable comments. We have replied to the questions in the above responses, including:
>
> - Expanded literature review;
> - Added two strong baselines for comparison;
> - Clarification on performance gains and training efficiency;
> - The relationship between LATR and MCTS.
>
> We hope the above response could resolve your concerns. If there are further questions, please do not hestitate to let us know. We are truly grateful for your contributions.

---

### Official Review · Reviewer_mqHB · 2025-10-31

**Soundness:** 3
**Presentation:** 3
**Contribution:** 3
**Rating:** 2
**Confidence:** 4

**Summary:**

The paper presents LATR: Lookahead Tree-based rollouts for improving the diversity of the rollouts in algorithms like GRPO.
LATR operates in 3 stages: Branching which creates new trajectories when the model is highly uncertain, (2) Lookahead simulation to extend a bnrach and Pruning when branches that are similar are pruned.

Branching occurs at each point with the highest logit-based token being extended to capture the most likely trajectory. To encoureage diversity, if other logits satisfy a threshold, child branches are created. The authors use dual-thresholds to prevent diverging too far from the models distribution.

Since it is computationally expensive to branch at each logit/token, the authors uses lookahead simulation. After a branch, each trajectory along the branch is generated sequentially for a window or r tokens. Finally, pruning is performed for these sequential generations so that branches that are not sufficiently different are pruned.

The authors explore more optimizations like early stopping and hybrid rollouts akin to epsilon in RL settings.

The authors then conduct an empirical evaluation and compare policy optimization methods with LATR and compare to stochastic sampling. The authors also provide ablations to contrast the effect of different components.

**Strengths:**

1. The paper is clear and well-written

2. The proposed technique is intuitive and explained well. Using tree-based methods to focus on diverse trajectories based on model certainty is intuitive.

3. The experiments showcase model improvements in performance but significant improvements in the inference costs.

**Weaknesses:**

1. I think one major weakness in the paper is the empirical evaluation which is my primary reason for a lower score.

There is only 1 model considered. I think to better understand the performance of LATR you would need to evaluate more models.

2. It is not clear how statistically significant the results are since standard deviations are not provided. How many times were the models trained for these experiments.

3. Other relevant baselines like TreeRL are not considered. I believe TreeRL also seeks to improve diversity by employing entropy.

4. What is the threshold for EditDistance used in the experiments?

5. Im not sure how useful Ablation 5.3 is. You are comparing two different methods. While LATR is not sensitive to temperature and this is not super surprising, the interesting analysis IMO is to showcase the sensitivity to all these new hypermeters/thresholds you have introduced.

**Questions:**

Overall, this paper is interesting and the results seem okay.  There are a few weaknesses that the authors can clarify before I can increase my score. Happy to discuss further.

---

> ### Author Response · Authors · 2025-11-21
> **Response to Reviewer mqHB [1/3]**
>
> Thank you for your appreciation of our clear writing, well-explained method, and significant improvements. For the concerns, we detail our response below. We would appreciate it if you could let us know whether your concerns are addressed.
>
> $~$
>
> ---
>
> $~$
>
> > ***Q1: I think one major weakness in the paper is the empirical evaluation which is my primary reason for a lower score. There is only 1 model considered. I think to better understand the performance of LATR you would need to evaluate more models.***
>
> **A1:** To address your concern and validate the generalizability of LATR beyond a single base model, we expand our empirical evaluation to include two additional models: Qwen2.5-7B and Qwen3-1.7B-Base. Using the Countdown dataset and the DAPO policy-update algorithm, we compare LATR with standard stochastic sampling across all three models. As shown in Table 1,  **LATR consistently outperforms the baseline on every model** , indicating that its effectiveness is not tied to a particular architecture or scale. These findings support the robustness and broad applicability of LATR.
>
> Table 1
>
> | Method                   | Pass@1 | Pass@8 |
> | ------------------------ | ------ | ------ |
> | Qwen2.5-3B               | 1.1    | 5.5    |
> | Qwen2.5-3B + Stoch.          | 70.7    | 78.0    |
> | Qwen2.5-3B + LATR          | 74.7    | 81.5    |
> | Qwen2.5-7B               | 1.1    | 5.4    |
> | Qwen2.5-7B + Stoch.      | 70.9   | 79.8   |
> | Qwen2.5-7B + LATR        | 76.0   | 82.1   |
> | Qwen3-1.7B-Base          | 2.0    | 9.5    |
> | Qwen3-1.7B-Base + Stoch. | 66.0   | 75.5   |
> | Qwen3-1.7B-Base + LATR   | 67.8   | 77.6   |
>
> $~$
>
> ---
>
> $~$
>
> > ***Q2: It is not clear how statistically significant the results are since standard deviations are not provided. How many times were the models trained for these experiments.***
>
> **A2:** Due to the substantial computational cost of RLVR training on large language models, it is common practice in the field to report results from a single training run per configuration. This is consistent with prior influential works such as DAPO [1] and ProRL [2], which also do not provide repeated runs or standard deviations.
>
> Nevertheless, to assess statistical significance, we conduct two additionaltraining runs on the Countdown dataset using the DAPO policy-update algorithm, yielding a total of three independent runs. The results (shown in Table 2) demonstrate that LATR consistently outperforms DAPO with stochastic sampling (70.7 for Pass@1 and 78.0 for Pass@8). A one-sided t-test yields **p-values of 1.8e-13** for Pass@1 and **1.7e-8** for Pass@8, confirming that the improvement in correctness is statistically significant.
>
> Table 2
>
> |   | Pass@1 | Pass@8 |
> | ------- | ------ | ------ |
> | Run #1  | 74.7   | 81.5   |
> | Run #2  | 75.3   | 82.4   |
> | Run #3  | 74.2   | 81.1   |
> | Mean    | 74.7   | 81.7   |
> | Std.    | 0.55   | 0.67   |
>
> [1] DAPO: An Open-Source LLM Reinforcement Learning System at Scale
>
> [2] ProRL: Prolonged Reinforcement Learning Expands Reasoning Boundaries in Large Language Models

---

> ### Author Response · Authors · 2025-11-21
> **Response to Reviewer mqHB [2/3]**
>
> > ***Q3: Other relevant baselines like TreeRL are not considered. I believe TreeRL also seeks to improve diversity by employing entropy.***
>
> **A3:** We thank the reviewer for raising this important point, and we have added the results and analyses to Appendix C.3 in the revised version.
>
> **[Mechanism Difference]** TreeRL indeed enhances diversity through its EPTree rollout strategy, which re-generates continuations at high-entropy tokens, an intuitive approach that leverages model uncertainty to spawn alternative branches. In contrast, LATR takes a fundamentally different approach: rather than relying solely on  **token-level entropy** , it uses lookahead simulation and similarity-guided pruning to explicitly promote  **trajectory-level divergence**. This allows LATR to explore a broader and more structurally diverse set of reasoning paths, while avoiding redundant or near-duplicate trajectories.
>
> **[Empirical Comparison]** To provide a direct empirical comparison, we evaluate both EPTree and LATR on the Countdown dataset under GRPO and DAPO. As summarized in Table 3,  **LATR achieves large and consistent improvements over stochastic sampling, whereas EPTree does not produce comparable gains** . This finding suggests that the strong performance reported in TreeRL stems primarily from its enriched process rewards for policy update rather than its rollout strategy. Specifically, TreeRL constructs a static tree during rollout, which enables backward propagation of sparse binary rewards from leaf nodes to internal nodes, thereby generating dense process-level rewards and advantages for policy update. In contrast, LATR improves RL performance by enhancing trajectory-level diversity without requiring modifications to the underlying policy update algorithm, highlighting its compatibility and effectiveness as a plug-in rollout strategy for existing RL frameworks.
>
> Table 3
>
> | Method          | Pass@1 | Pass@8 | Avg.Len (Pass@1) | Avg.Len (Pass@8) |
> | ------------- | ------------------------- | ------------------------- | ------------------------ | ------------------------ |
> | GRPO w Stoch. | 65.9                      | 73.9                      | 473                      | 610                      |
> | GRPO w EPTree | 65.3                      | 73.5                      | 471                      | 599                      |
> | GRPO w LATR   | **70.9**            | **77.4**            | **378**                 | **469**            |
> | DAPO w Stoch. | 70.7                      | 78.0                      | 483                      | 630                      |
> | DAPO w EPTree | 66.3                      | 74.6                      | 450                      | 595                      |
> | DAPO w LATR   | **74.7**            | **81.5**            | **367**                 | **453**            |

---

> ### Author Response · Authors · 2025-11-21
> **Response to Reviewer mqHB [3/3]**
>
> > ***Q4: What is the threshold for EditDistance used in the experiments?***
>
> **A4:** As described in Appendix B.3, we use $\tau_{ed}=0.4$ for all the experiments. We will move these configurations from the appendix into the main text in the revised manuscript for clarity.
>
>
> $~$
>
> ---
>
> $~$
>
>
> > ***Q5: Im not sure how useful Ablation 5.3 is. You are comparing two different methods. While LATR is not sensitive to temperature and this is not super surprising, the interesting analysis IMO is to showcase the sensitivity to all these new hypermeters/thresholds you have introduced.***
>
> **A5:** In Ablation 5.3, our goal is to isolate how LATR behaves under different **sampling temperatures** while keeping the pruning thresholds $\tau_{\text{abs}}$ and $\tau_{\text{rel}}$ fixed. As discussed in ProRL [2], increasing the temperature increases token-level diversity and can indirectly enlarge trajectory-level diversity. Our experiment is designed to illustrate that (1)  **LATR consistently improves performance across a wide range of token-level diversities** , and (2)  **the method remains robust even under excessively high temperatures** , where sampling becomes notably unstable.
>
> **[Additional Hyper-parameter Experiments]** We agree with the reviewer that examining sensitivity to explicit hyperparameters is also valuable. To this end, we conduct additional experiments varying the pruning threshold $\tau_{ed} \in \{0.3, 0.4, 0.5, 0.6\}$. As summarized in the Table 4 (and included in Appendix C.5 of the revised version), the results show that LATR maintains stable performance across this range, indicating that the method is robust to reasonable choices of pruning thresholds. We are still running the experiments with the branching thresholds $\tau_{abs}, \tau_{rel}$, which are expected to complete in a few days. We will further modify the manuscript to include these new results.
>
> Table 4
>
> | $\tau_{ed}$ | Pass@1 | Pass@8 |
> | ------------- | ------ | ------ |
> | 0.3           | 73.8   | 80.5   |
> | 0.4           | 74.7   | 81.5   |
> | 0.5           | 74.1   | 81.4   |
> | 0.6           | 73.4   | 79.8   |

---

> > ### Comment · Reviewer_mqHB · 2025-11-25
> >
> > Thanks for your response. The rebuttal has addressed all my concerns and I will raise my score to a 6.
> >
> > I think for the final version, the paper's impact would be improved if you consider even more (and larger) models like those with 30B or more parameters.

---

> > > ### Author Response · Authors · 2025-11-26
> > > **Thanks for your response**
> > >
> > > Thank you for your prompt response. We are delighted that our rebuttal has addressed your concerns, and we sincerely appreciate your positive assessment of our work. Thanks as well for the valuable suggestion about testing larger models, which is an important next step for future work to further validate our approach at scale.

---

### Official Review · Reviewer_cvuy · 2025-10-31

**Soundness:** 3
**Presentation:** 3
**Contribution:** 3
**Rating:** 8
**Confidence:** 3

**Summary:**

This paper introduces an enhancement to the reinforcement learning with verifiable rewards algorithm.

Motivation

Previous existing methods for RLVR, such as GRPO and DAPO, suffer from limited diversity in the sampled trajectories. Follow-up works mitigate this by (i) including increasing sampling temperature or (ii) filtering out and rejecting sequences with very similar samples. However the authors argue that  (i) i focuses on token-level diversity, rather than trajectory-leval diversity, and (ii) needs multiple rollouts leading to high cost, and  lacks withing-group diversity.

Method

To mitigate this, the authors propose  Lookahead Tree-Based Rollout (LATR) algorithm, inspired by MCTS that works as follows:
starting from a root branch that corresponds to the prompt:
- first there is a branching stage, where each branch is extended with its highest-probability token (this is the parent branch) , as well as other tokens that have a "high-enough probability"  (these are the children branches), This is done until we have a set of K sequences.
- then, there is a pruning stage: given a lookahead step r , they look at each children branch that already had r rollouts from their parent branch, and they look at how different this children branch is from the parent branch (using edit distance). If differrent enough, the children branch and its descendants are kept. If not, it is removed along with its descendants.

They introduce further optimizations eg.. slowly decreasing the ratio of sequences generated with LATR during training in favor of standard stochastic sampling (more exploration in the start, more exploitation in the end of training)

Experiments

They run experiments on the CountDown,  DAPO-math, ACM23, MATH-500, Olympiad-Bench datasets. They run GRPO w LATR / DAPO w LATR, and they compare them to the baselines od GRPO w stochastic sampling / DAPO w stochastic sampling. Performance is measured in terms of (i) average correctness of the provided answer and (ii) average length of the answer provided (as well as format for CountDown). In almost all cases, LATR shows better performance in both metrics compared to the baselines. Taining dynamic results reveal faster learning (in terms of avg val reward) with LATR.

They also run ablation studies on the diversity comparison across answers with / wo LATR, the effect of the pruning, and of sampling at different temperatures.

**Strengths:**

- Good presentation of the paper, clarity and ease of reading
- Good contextualization of the paper in previous work, helpful for those who are not experts in this field
- Simplicity of the introduced changes and strong evidence that it yields improvements
- Clear experiments section with ablation studies

**Weaknesses:**

- I think an important ablation study is missing to show the claim of the paper that trajectory level lookahead is important and yields improvements over token-level lookahead.
- Also, the authors mention that "token-level variations typically occur without lookahead ability, making local deviations (e.g., substituting “compute” with “calculate”)  " however this can also happen in trajectory level variations (with multiple such subsitutions). The authors use the edit distance to quantity diversity, however this distance assigns high value to word substitutions that mean the same thing.

**Questions:**

- Given point 1 above, Could you show ablation studies on the lookahead step r, in particular with r = 1 (i.e. only one-step lookahead but with the pruning that removed redundancy ) ? I would be curious to know if trajectory-level lookahead is important, or if token-level lookahead with diversity pruning is enough.
- Given point 2 above, do you think edit distance a good measure of diversity between sequences? You could have 2 sequences that have high edit distance but mean the same thing because there are synonym words. Have you thought of using another distance e.g.  distance between the embeddings of the sequence, to make sure that they are semantically different?
- In the diversity experiments, "two answer expressions ...  if their evaluated numerical outcomes differ". I am confused: for each math question there is only one correct answer, and you can have multiple diverse ways to reach this correct answer. Why aren't you looking at this diversity instead (rather than diversity over final answers, for which high diversity means that some answers are incorrect! )
- For the temperature-ablation experiments, do you keep the same theresholds tau_abs and tau_rel across temperatures? If yes, could you elaborate why is LATR less sensitive to temperature tuning? The branching quality in LATR will still be affected by the increase in temperature (because of the increased stochasticity in the new branches ), no?

---

> ### Author Response · Authors · 2025-11-21
> **Response to Reviewer cvuy [1/3]**
>
> Thank you for your appreciation of our good presentation, beneficial methods, and clear experiments. For the concerns, we detail our response below. We would appreciate it if you could let us know whether your concerns are addressed.
>
> > ***Q1: I think an important ablation study is missing to show the claim of the paper that trajectory level lookahead is important and yields improvements over token-level lookahead. Could you show ablation studies on the lookahead step r, in particular with r = 1 (i.e. only one-step lookahead but with the pruning that removed redundancy ) ? I would be curious to know if trajectory-level lookahead is important, or if token-level lookahead with diversity pruning is enough.***
>
> **A1:** In our branching mechanism, each branch is initiated by a distinct first token. Therefore, token-level lookahead corresponds to the case $r=2$: the model observes **only the second token** in each branch before deciding whether to prune. To address the reviewer’s question, we conduct an ablation where pruning decisions is based exclusively on the second token, i.e., a branch is pruned if its next token matches that of another branch. Results are reported in Table 1.
>
> Table 1
>
> | Method                       | Pass@1 | Pass@8 |
> | ---------------------------- | ------ | ------ |
> | Stochastic                   | 70.7   | 78.0   |
> | LATR w rand branch           | 69.6   | 75.8   |
> | LATR w rand prune            | 72.5   | 79.2   |
> | LATR w/o prune               | 71.0   | 78.7   |
> | LATR w token-level lookahead | 72.1   | 80.4   |
> | LATR                         | 74.7   | 81.5   |
>
> While this token-level lookahead variant outperforms both Stochastic Sampling and the no-pruning baseline, it still falls short of the full LATR strategy. This gap arises because token-level lookahead is limited to capture trajectory divergence. Pruning decisions made from only the next token often fails to predict later divergence, causing the premature removal of potentially valuable branches and degrading rollout quality. These findings confirm that  **trajectory-level lookahead is essential** .

---

> ### Author Response · Authors · 2025-11-21
> **Response to Reviewer cvuy [2/3]**
>
> > ***Q2: Do you think edit distance a good measure of diversity between sequences? You could have 2 sequences that have high edit distance but mean the same thing because there are synonym words. Have you thought of using another distance e.g. distance between the embeddings of the sequence, to make sure that they are semantically different?***
>
> **A2:** Thank you for raising this question. Our similarity metric is not intended to measure general semantic equivalence. Instead, its role is to compare different partial rollout trajectories for the same question, in order to determine whether two prefixes are likely to lead to **distinct exploration strategies or  different final answers**.
>
> We initially explored embedding-based semantic similarity (using Qwen3-Embedding-0.6B) as an alternative to Edit Distance. However, **embedding-based approaches are ineffective for distinguishing the fine-grained logical differences** that frequently arise when multiple solution paths are explored for a single problem. Specfically in mathematical and logical reasoning, different paths tend to diverge at crucial tokens, typically numbers, operators, or short logical cues. Embedding models map such tokens to very similar vectors, making sentence-level representations nearly indistinguishable even when the underlying reasoning differs substantially. As shown in Table 2, trajectory pairs that clearly diverge and those that do not both yield cosine similarities around 0.92, demonstrating that embedding-based similarity is insensitive to the reasoning distinctions we aim to capture.
>
> Table 2
>
> | Sequences                                                                                                                                                                                                                      | Diverge in reasoning? | Cosine similarity | Edit Distance |
> | ------------------------------------------------------------------------------------------------------------------------------------------------------------------------------------------------------------------------------ | --------------------- | ----------------- | ------------- |
> | (1) First, maybe I can try addition and subtraction. 60 - 51 is 9. Then 35 - 4 is 31. 9 + 31 is 40. Not enough. (2) First, maybe I can try addition and subtraction. 60 - 35 is 25. Then 51 - 25 is 26. 26 + 4 is 30. Not enough. | yes                   | 0.92              | 0.3125        |
> | (1) Let’s try multiplying the previous numbers. 25 * 4 = 100. 100 - 51 = 49. Got it! (2) Maybe I should consider multiplying. 25 * 4 = 100. 100 - 51 = 49. Got it!                                                               | no                    | 0.92              | 0.2           |
>
> In contrast, we observe that when two trajectories diverge in reasoning logic, this  **divergence typically appears as token-level differences that persist over a sufficiently long lookahead window**; when trajectories converge, their continuations rapidly align as autoregressive models are highly sensitive to input prefixes. Edit Distance therefore serves as a practical and reliable proxy for detecting reasoning divergence, whereas embedding-based metrics blur these differences.
>
> To further validate our choice, we conduct additional experiments comparing the embedding-based metric and two alternative lightweight token-level metrics, including (1) ROUGE-L: which measures the length of the longest common subsequence; (2) Suffix matching: which measures the length of the longest suffix of one sequence that appears anywhere in the other.
>
> Table 3
>
> | Method          | Pass@1 | Pass@8 | Avg.Len (Pass@1) | Avg.Len (Pass@8) |
> | --------------- | ------------------------- | ------------------------- | ------------------------ | ------------------------ |
> | Edit Distance   | 74.7                      | 81.5                      | 367                      | 453                      |
> | ROUGE-L         | 73.9                      | 80.5                      | 390                      | 486                      |
> | Suffix Matching | 74.9                      | 81.7                      | 388                      | 493                      |
> | Embedding-based | 72.9                      | 79.8                      | 369                      | 445                      |
>
> As shown in Table 3, embedding-based similarity yields the weakest performance, whereas all token-level metrics achieve comparable and substantially higher final accuracy. Given its simplicity and competitive effectiveness, we retain Edit Distance as our pruning criterion. We have expanded this analysis the revised manuscript (Appendix C.4) and revised the term “semantic similarity” to avoid potential misinterpretation.

---

> ### Author Response · Authors · 2025-11-21
> **Response to Reviewer cvuy [3/3]**
>
> > ***Q3: In the diversity experiments, "two answer expressions ... if their evaluated numerical outcomes differ". I am confused: for each math question there is only one correct answer, and you can have multiple diverse ways to reach this correct answer. Why aren't you looking at this diversity instead (rather than diversity over final answers, for which high diversity means that some answers are incorrect! )***
>
> **A3:** We acknowledge that "diversity" can refer to two distinct notions: (1)  **Path diversity** : different reasoning trajectories that converge to the same (ideally correct) answer; and (2)  **Outcome diversity** : trajectories that differ both in reasoning paths and final answers (some of which may be incorrect).
>
> In the context of RL, **outcome diversity is essential to policy refinement.** First, it introduces high-variance rewards by mixing correct and incorrect outcomes, enriching the learning signal for policy updates. Second, recent work [1] demonstrates that negative samples (i.e., incorrect answers) play a pivotal role in RL training: they help the policy learn to avoid implausible reasoning paths without stifling exploration. Thus, while path diversity is valuable, outcome diversity provides the critical signal needed for effective policy refinement in our setting.
>
> [1] The Surprising Effectiveness of Negative Reinforcement in LLM Reasoning.
>
> $~$
>
> ---
>
> $~$
>
> > ***Q4: For the temperature-ablation experiments, do you keep the same theresholds tau_abs and tau_rel across temperatures? If yes, could you elaborate why is LATR less sensitive to temperature tuning? The branching quality in LATR will still be affected by the increase in temperature (because of the increased stochasticity in the new branches ), no?***
>
> **A4:** Yes, we keep the thresholds $\tau_{abs}$ and $\tau_{rel}$ fixed across different temperatures. Although temperature reshapes the next-token distribution, LATR is less sensitive to this change due to its **threshold-guided branching** mechanism, which naturally filters out low-likelihood continuations.
>
> In standard Stochastic Sampling, higher temperatures flatten the next-token distribution, increasing the probability of sampling very low-likelihood tokens. Training on such low-likelihood trajectories can destabilize learning and even lead to policy collapse. In contrast, LATR deterministically retains only those candidate tokens whose probabilities exceed $\tau_{abs}$. This filtering mechanism ensures that, even under distributions reshaped by high temperature, LATR avoids incorporating extremely unlikely tokens into rollouts. As a result, LATR maintains more stable and on-policy training trajectories, making it robust to temperature variations.

---

> ### Author Response · Authors · 2025-11-27
> **We look forward to your response**
>
> Thanks again for your valuable comments. We have replied to the questions in the above responses, including:
>
> - Ablation study on token-level lookahead;
> - Explanation and comparison of similarity measures used in pruning;
> - Clarification on diversity evaluations;
> - Clarification on sensitivity to temperature.
>
> We hope the above response could resolve your concerns. If there are further questions, please do not hestitate to let us know. We are truly grateful for your contributions.

---

### Official Review · Reviewer_gVkW · 2025-11-03

**Soundness:** 3
**Presentation:** 3
**Contribution:** 2
**Rating:** 6
**Confidence:** 3

**Summary:**

This paper introduces a tree-based algorithm to increase the sampling diversity in the setting of RLVR.  The method focuses in generating trajectory-level diversity during the sampling process by rolling out and pruning semantically similar trajectories. The show that including this sampling strategy with GRPO and DAPO, the performance increases given the new variety of sampled reasoning trajectories. They evaluate their method on math and logical reasoning tasks.

**Strengths:**

- The paper is generally well written and the method is explained clearly.
- The empirical evaluation shows the benefits of the enhanced strategy with respect to token-level stochastic sampling

**Weaknesses:**

- The paper claims to care about semantic similarity but they prune based on Edit distance which doesn’t seem to me to be a good measure of semantic similarity. Maybe could the authors explain why this is working in their evaluation tasks?
- It seems that the new sampling process introduces could potentially introduce off-policy issues? Is this being taken into account in $\pi_{old}$, or am I misunderstanding  something?

**Questions:**

See above.

---

> ### Author Response · Authors · 2025-11-21
> **Response to Reviewer gVkW [1/2]**
>
> Thank you for your appreciation of our clear writing and beneficial methods. For the concerns, we detail our response below. We would appreciate it if you could let us know whether your concerns are addressed.
>
> $~$
>
> ---
>
> $~$
>
> > ***Q1: The paper claims to care about semantic similarity but they prune based on Edit distance which doesn’t seem to me to be a good measure of semantic similarity. Maybe could the authors explain why this is working in their evaluation tasks?***
>
> **A1:** Thank you for raising this question. Our similarity metric is not intended to measure general semantic equivalence. Instead, its role is to compare different partial rollout trajectories for the same question, in order to determine whether two prefixes are likely to lead to **distinct exploration strategies or different final answers**.
>
> We initially explored embedding-based semantic similarity (using Qwen3-Embedding-0.6B) as an alternative to Edit Distance. However, **embedding-based approaches are ineffective for distinguishing the fine-grained logical differences** that frequently arise when multiple solution paths are explored for a single problem. Specfically in mathematical and logical reasoning, different paths tend to diverge at crucial tokens, typically numbers, operators, or short logical cues. Embedding models map such tokens to very similar vectors, making sentence-level representations nearly indistinguishable even when the underlying reasoning differs substantially. As shown in Table 1, trajectory pairs that clearly diverge and those that do not both yield cosine similarities around 0.92, demonstrating that embedding-based similarity is insensitive to the reasoning distinctions we aim to capture.
>
> Table 1
>
> | Sequences                                                                                                                                                                                                                      | Diverge in reasoning? | Cosine similarity | Edit Distance |
> | ------------------------------------------------------------------------------------------------------------------------------------------------------------------------------------------------------------------------------ | --------------------- | ----------------- | ------------- |
> | (1) First, maybe I can try addition and subtraction. 60 - 51 is 9. Then 35 - 4 is 31. 9 + 31 is 40. Not enough. (2) First, maybe I can try addition and subtraction. 60 - 35 is 25. Then 51 - 25 is 26. 26 + 4 is 30. Not enough. | yes                   | 0.92              | 0.3125        |
> | (1) Let’s try multiplying the previous numbers. 25 * 4 = 100. 100 - 51 = 49. Got it! (2) Maybe I should consider multiplying. 25 * 4 = 100. 100 - 51 = 49. Got it!                                                               | no                    | 0.92              | 0.2           |
>
> In contrast, we observe that when two trajectories diverge in reasoning logic, this  **divergence typically appears as token-level differences that persist over a sufficiently long lookahead window**; when trajectories converge, their continuations rapidly align as autoregressive models are highly sensitive to input prefixes. Edit Distance therefore serves as a practical and reliable proxy for detecting reasoning divergence, whereas embedding-based metrics blur these differences.
>
> To further validate our choice, we conduct additional experiments comparing the embedding-based metric and two alternative lightweight token-level metrics, including (1) ROUGE-L: which measures the length of the longest common subsequence; (2) Suffix matching: which measures the length of the longest suffix of one sequence that appears anywhere in the other.
>
> Table 2
>
> | Method          | Pass@1 | Pass@8 | Avg.Len (Pass@1) | Avg.Len (Pass@8) |
> | --------------- | ------------------------- | ------------------------- | ------------------------ | ------------------------ |
> | Edit Distance   | 74.7                      | 81.5                      | 367                      | 453                      |
> | ROUGE-L         | 73.9                      | 80.5                      | 390                      | 486                      |
> | Suffix Matching | 74.9                      | 81.7                      | 388                      | 493                      |
> | Embedding-based | 72.9                      | 79.8                      | 369                      | 445                      |
>
> As shown in Table 2, embedding-based similarity yields the weakest performance, whereas all token-level metrics achieve comparable and substantially higher final accuracy. Given its simplicity and competitive effectiveness, we retain Edit Distance as our pruning criterion. We have expanded this analysis the revised manuscript (Appendix C.4) and revised the term “semantic similarity” to avoid potential misinterpretation.

---

> ### Author Response · Authors · 2025-11-21
> **Response to Reviewer gVkW [2/2]**
>
> > ***Q2: It seems that the new sampling process introduces could potentially introduce off-policy issues? Is this being taken into account in $\pi_{old}$, or am I misunderstanding something?***
>
> **A2:** In the GRPO algorithm (including its successors such as DAPO and GSPO), all trajectory samples are generated according to the next-token probability distribution strictly from the policy $\pi_{old}$, and both stochastic sampling baseline and LATR follow this principle. The key difference lies in how tokens are selected. Standard stochastic sampling may occasionally sample low-probability tokens from the next-token distribution, while LATR applies deterministic filtering based on the absolute and relative thresholds $\tau_{abs}$ and $\tau_{rel}$ to retain only high-probability candidates. As a result, the rollout trajectories generated by LATR typically have a higher cumulative likelihood under $\pi_{old}$ than trajectories obtained via stochastic sampling. From this perspective,  **LATR does not introduce additional off-policy bias but actually reduces the off-policy risk because its trajectories stay closer to the high-probability regions of $\pi_{old}$**.

---

> ### Author Response · Authors · 2025-11-27
> **We look forward to your response**
>
> Thanks again for your valuable comments. We have replied to the questions in the above responses, including:
>
> - Explanation and comparison of similarity measures used in pruning;
> - How LATR mitigates off-policy issues.
>
> We hope the above response could resolve your concerns. If there are further questions, please do not hestitate to let us know. We are truly grateful for your contributions.

---

### Author Response · Authors · 2025-11-21
**We sincerely thank the reviewers and AC**

We thank the reviewers for their constructive feedback and recognition of our work's significance, including insightful motivation (Vq12) beneficial methods (gVkW, cvuy), significant improvements (mqHB), and good presentation (gVkW, cvuy, mqHB, Vq12).

In response to reviewers' comments/questions, we addressed the following points in our rebuttal:

* Explanation on similarity measures used in pruning (gVkW, cvuy)
* Experiments on more base models (mqHB)
* Comparison with other rollout strategies (mqHB, Vq12)
* Additional ablation studies, including token-level lookahead (cvuy) and branching/pruning thresholds (mqHB)
* Verification on statistical significance (mqHB)
* Clarification on sensitivity to temperature and how LATR mitigates off-policy issues (gVkW, cvuy)
* Clarification on diversity evaluations (cvuy)
* Complimentary literature review, clarification on performance and efficiency, and Explanation on the relationship with MCTS (Vq12)

We hope these responses address your key concerns! We welcome any further feedback and are happy to make additional edits as needed.

---

### Author Response · Authors · 2025-12-02
**Summary of Rebuttal**

Dear Area Chair,

We sincerely appreciate the time and effort you and the reviewers have dedicated to the review process. We are writing to provide a brief summary of the rebuttal to assist you in your final evaluation.

First, we are grateful for the reviewers' recognition of our work's strengths, including **insightful motivation** (`Vq12`), **beneficial methods** (`gVkW`, `cvuy`), **significant improvements** (`mqHB`), and **good presentation** (`gVkW`, `cvuy`, `mqHB`, `Vq12`).

During the rebuttal period, we engaged constructively with the reviewers’ concerns. In particular, we thank reviewer `mqHB` for their prompt response, in which they stated: “The rebuttal has addressed all my concerns and I will raise my score to a 6.” This message was sent on 25 November 2025 at 01:28 UTC, well **before the reported leakage event** on 27 November 2025 at 15:00 UTC.

Although the other three reviewers did not respond in time, we have nonetheless provided extensive additional experiments, ablation studies, and detailed clarifications in our rebuttal to address each of their concerns point-by-point. We note that reviewer `Vq12`, who assigned the only negative score `4` currently (taken into account the updated estimation by `mqHB`) , also indicated low confidence `2`, which we hope will be taken into account during the final evaluation.

We hope this summary aids your synthesis and final decision. Thank you again for your dedication to the thorough review process.

Sincerely,

The Authors

---

### Meta-Review · Area_Chair_PXqM · 2026-01-08

**Summary:**

This submission focused on the rollout diversity dilemma during the standard token-level stochastic sampling, and proposed a lookahead tree-based rollout sampling method (LATR) to promote trajectory-level diversity. By designing the proper branching, lookahead simulation and then pruning, the proposed method can effectively accelerate the policy learning and improve the final performance on GRPO and DAPO across a range of reasoning tasks.

**Reviewer Concerns:**

The reviewers' concerns can be summarized as the following points:

1) About the design, the paper claims to promote semantic diversity, yet relies heavily on edit distance for pruning and diversity measurement, which poorly captures semantic similarity and over-penalizes synonymous substitutions. The use of edit distance raises concerns about whether the observed benefits truly reflect semantic diversity, and alternative measures (e.g., embedding-based similarity) or stronger justification are needed.

2) Insufficient Ablations. Key claims—such as the necessity of trajectory-level lookahead over token-level lookahead—are not adequately supported by ablation studies. In particular, experiments with minimal lookahead (e.g., r=1) combined with redundancy pruning are missing, making it unclear whether the gains stem from deeper lookahead or from simpler diversity mechanisms.

3) For the main experiments, the evaluation is limited to a single model, lacks standard deviations or multiple runs, and does not assess statistical significance. Performance gains are modest, correctness improvements are limited, and several relevant baselines (e.g., TreeRL, entropy-based diversity methods) are missing, weakening the empirical evidence.

4) Some points are not clear. Specifically, the new sampling process may introduce off-policy effects, which are not clearly addressed. Several algorithmic details are underspecified (e.g., edit-distance thresholds, temperature interactions, pruning hyperparameters), and the analogy to MCTS is underexplained, making it difficult to assess the method’s theoretical grounding and novelty. Besides, the literature review omits many related lookahead-based LLM approaches, and the baselines used are not state-of-the-art. While inference cost is discussed, training cost comparisons are missing, despite the likelihood that lookahead methods incur higher training overhead. Overall, the positioning relative to prior work and the practical cost–benefit tradeoff remain unclear.

**Reviewer Scores:**

The reviewers rated the submission with the scores 6, 8, 2, 4 respectively at the initial stage.

After the first-round response, the third reviewer who rated 2 claimed to increase the score to 6, considering the substantial clarification and further verification during the rebuttal. Note that, the score increasing happens before the reported bug date (although it is the exact date). Based on the detailed and clear response of the author regarding the concerns, AC would like to consider it reliable.

Regarding the comments of the reviewer who rated 4, the authors have well revised the submission to include the discussion about lookahead reasoning methods, and add more comparison to demonstrate the performance. Besides, after checking the performance gain limitation, it seems that the improvement of 3%-4% is decent as clarified by the authors. AC thinks if the reviewer can participate fully in the discussion, he/she may improve the score.

---

### Decision · Program_Chairs · 2026-01-26

Accept (Poster)